



# Intercomparison of AOD retrievals from GAW-PFR and SKYNET sun photometer networks and the effect of calibration

Angelos Karanikolas[1,2], Natalia Kouremeti[1], Monica Campanelli[3], Victor Estellés[4,3], Masahiro Momoi[5], Gaurav Kumar[4] and Stelios Kazadzis[1]

[1] World Optical Depth Research and Calibration Centre (WORCC), Physikalisch-Meteorologisches Observatorium Davos/World Radiation Center (PMOD/WRC), Davos Dorf, 7260, Switzerland
[2] Institute for Particle Physics and Astrophysics, ETH Zurich, Zurich, 8093, Switzerland
[3] Institute of Atmospheric Sciences and Climate (ISAC), Consiglio Nazionale Delle Ricerche (CNR), Rome, 00133, Italy
[4] Earth Physics and Thermodynamics Department, Universitat de València, Valencia, 46100, Spain
[5] GRASP SAS, Lezennes, 59260, France

*Correspondence to*: Angelos Karanikolas (angelos.karanikolas@pmodwrc.ch)

**Abstract.** In this study, we assess the homogeneity of aerosol optical depth (AOD) between the sun photometer networks Global Atmospheric Watch-Precision Filter Radiometer (GAW-PFR) and European Skynet Radiometers network (ESR) at the 2 common wavelengths of their main instruments 500 nm and 870 nm. The main focus of the work is on evaluating the effect of the Improved Langley calibration method, (ILP) used by SKYNET and investigating the factors affecting its performance. We used data from three intercomparison campaigns that took place in the period 2017-2021. Each campaign has two phases in two locations. One is mountainous rural (Davos, Switzerland) and the other urban (Rome, Italy). Our analysis shows that the AOD differences due to post processing and instrument differences are minor. The major factor leading to AOD differences is the calibration method where we found a systematic underestimation of AOD compared to GAW-PFR due to an underestimation in the ILP calibration. The calibration and AOD differences are smaller in Davos where at 870 nm the traceability criteria are satisfied and at 500 nm the median differences are below 0.01. In Rome at 500 nm the AOD median differences per campaign are between 0.015-0.035. Attempting to explain the differences we found no association between the calibration performance and the level or the variability of the aerosol properties. We also conducted a sensitivity study, which shows that part of the difference can be potentially explained by errors in the assumed surface albedo and instrument solid view angle provided as inputs to the ILP code (based on Skyrad pack 4.2). Our findings suggest that the ILP method is mainly sensitive to the measured sky radiance. The calibration underestimation is probably caused by an error on the retrieved scattering aerosol optical depth (sc-AOD) through the sky radiance inversion. Using an alternative retrieval method (Skyrad MRI pack version 2) to derive sc-AOD and repeat ILP calibration, we found no significant differences between the retrieved sc-AOD nor systematic increase of the calibrations. The potential error may be a result of the forward model assumptions, To conclude, calibration of sun photometers on site offers the advantage of avoiding instrument shipments and data gaps. However, ILP shows larger uncertainty and significant systematic difference compared to the traditional Langley calibration performed under low and constant AOD conditions at high altitude sites, due to the uncertainties of the calibration method and the input parameters needed for it. In the following sections we report on results on the AOD retrievals of several instruments





in different environments using different principles in their calibration methods. We also perform an investigation to explain
the causes of differences.

## 1 Introduction

Atmospheric particulate matter (aerosols) is a component of high importance in atmospheric sciences and modern environmental problems. They scatter and absorb solar radiation significantly affecting the Earth's energy budget. They also greatly assist water and ice nucleation in the atmosphere leading to the formation of clouds (Winkler & Wagner, 2022; Maloney et al., 2022). Aerosols were the major driver of surface solar radiation variations for several decades (Wild, 2012; et al., Correa et al., 2023). Affecting the surface solar radiation alters the exposure of organisms to biologically active radiation (Barnes et al., 2019; Bais et al., 2018) and solar energy production systems capabilities (Papachristopoulou et al., 2023; Hou et al., 2022). Both their direct and indirect effects on surface solar radiation are a significant forcings of the climate and remain the source of the largest uncertainty in radiative forcing attribution (IPCC, 2021).

According to the World Meteorological Organization (WMO), the most important parameter related to aerosols for Earth energy budget studies is the aerosol optical depth (AOD) (WMO, 2003). AOD describes the overall effect of the total aerosol column on solar radiation attenuation. AOD is an indicator of the total aerosol load in the atmosphere and its spectral dependence with the size of aerosols. AOD is calculated from direct solar irradiance (DSI) measurements by subtracting the effect of gas absorption and scattering at the absence of clouds covering the solar disk. The main instruments used for this purpose are the sun photometers. Sun photometers measure the DSI at selected wavelengths in which gas absorption is minimal and the AOD calculation can be more accurate.

There are different types of sun photometers used worldwide. There are several stations using the same type of sun photometer, which belong to an instrument network. The main sun photometer networks are the Aerosol Robotic Network (AERONET), Global Atmospheric Watch-Precision Filter Radiometer (GAW-PFR) and SKYNET. AERONET is the largest network with more than 400 stations worldwide and uses the CIMEL sun-sky photometer (hereafter CIMEL) as standard instrument (Holben et al., 1998). The GAW-PFR includes 15 stations mainly in remote worldwide locations. Its standard instrument is the Precision Filter Radiometer (PFR) and includes the WMO AOD reference instruments (PFR-Triad) (Kazadzis et al., 2018b). SKYNET is a multi-instrument research network divided in sub-networks and includes around 100 stations mainly in East Asia and Western Mediterranean regions. Its standard instrument for AOD observation is the PREDE-POM sun and sky radiometer (hereafter POM) (Nakajima et al., 2020). Each sub-network has developed its own calibration protocols and post processing algorithms independently. Especially, two procedures developed by sub-networks led by European Sky Radiometer network (ESR) and Center for Environmental Remote Sensing (CEReS) of Chiba University are recognized as the standard in the International Skynet Committee (Nakajima et al., 2020). Due to the differences among the networks (i.e., AERONET, GAW-PFR, SKYNET), it is important to evaluate the extent of homogeneity between the networks to ensure that the AOD observations are comparable and of similar accuracy. For this purpose, every 5 years the Filter Radiometer Comparison (FRC)





campaign takes place in Davos, Switzerland including instruments from all types (Kazadzis et al., 2023). There are several other intercomparison campaigns (Doppler et al., 2023), but also long-term comparisons between different networks (Cuevas et al, 2019; Karanikolas et al., 2022).

A necessary parameter for the AOD calculation is the DSI the instrument would measure at the top-of-the atmosphere

(extraterrestrial or calibration constant). There are different ways to calibrate a sun photometer. It can be accomplished either by using a co-located instrument as a reference, by laboratory calibration to the international system of units (SI) and use of satellite measurements for the top-of-the atmosphere or by using an indirect method to calibrate the instrument through the DSI at the ground. The conventionally used methods are the standard Langley plot method (SLP) (Shaw et al., 1973) and the calibration transfer from a reference instrument. Recent developments show that the laboratory calibration can also be accurate

(Gröbner & Kouremeti, 2019; Kouremeti et al.,2022; Gröbner et al., 2023). Another method is the improved Langley plot method (ILP) (Tanaka et al., 1986; Campanelli et al., 2004). This is a modification of SLP which accounts for AOD variations during the day in contrast to SLP that assumes AOD constant. The assumption of constant AOD results to larger error in more polluted areas hence SLP is applied only in high altitude locations. The aim of ILP is to calibrate instruments in the station they are normally operated regardless of the station's location instead of being transported to a calibration site. Therefore, this

method ideally brings the advantage to avoid damage in the transportation, missing data in the calibration period, low cost in the maintenance, and frequently tracking the variation of the calibration constant. AERONET and GAW-PFR calibrate the instruments either by SLP in Mauna Loa, Hawaii and Izaña, Tenerife or by calibration transfer from reference instruments, while SKYNET uses the ILP method.

Other than calibration procedure, each network also uses a different post processing and cloud screening algorithm to derive

from DSI and filter the AOD observations. One of the main differences are the inclusion of nitrogen dioxide ($NO_2$) and water vapor ($H_2O$) absorption from AERONET and SKYNET (Kazadzis et al., 2018a, Estellés et al., 2012, Drosoglou et al., 2023, Sinyuk et al., 2020), however there are differences in the way the optical depth of ozone absorption and Rayleigh scattering are calculated (Cuevas et al., 2019). The cloud screening algorithms also show some differences with the SKYNET algorithm being particularly strict (Kazadzis et al., 2018a).

In order to evaluate the ILP method World Optical Depth Research and Calibration Centre (WORCC) and European Skynet Radiometers network (ESR) have signed and Memorantum of Understanding (MoU) for scientific collaboration including several campaigns organized (Quality and Traceability of Atmospheric Aerosol Measurements or QUATRAM I, II and III). During the period 2017-2021 a PFR was transported to Sapienza University in Rome, Italy once for each campaign for several weeks or months to measure AOD in parallel with one or more POMs and CIMEL (Table 1). Also, at least one POM was

transported to Davos on 3 different periods as well (Table 1), where the WMO AOD reference (PFR-Triad) and a CIMEL are operated. The POMs were calibrated both with the ILP method and by calibration transfer using a PFR as a reference. There is already a publication under review showing calibration differences between several calibration methods (Campanelli et al., 2023).





This study aims to assess the AOD differences between GAW-PFR and ESR and the effect of the different calibration

approaches. Also, investigate the extent to which different factors such as the atmospheric conditions and the input parameters

required to perform ILP, contribute to the calibration and as a result to retrieved AOD differences.

## 2 Instruments and methods

### 2.1 Instrumentation and locations

The data used are from the period 2017-2021 in two locations, Davos (Switzerland) and Rome (Italy) in order to evaluate the

ILP performance under different conditions. The station of Davos is at PMOD/WRC 1590 m a.s.l. next to a town deep in the

Eastern Alps mountain range. The area has no significant local pollution. Aerosols can reach the area from other parts of

Europe due to its proximity with several European countries and during strong Sahara dust transport episodes. The other station

is in Rome at Sapienza University at 83 m a.s.l. close to the centre of Rome, the capital city of Italy.

For this study, we used the PFRN27 as reference in Davos (part of the PFR reference triad), while in Rome the PFRN14 (2017-

2019) and PFRN01 (2021). We also used the co-located CIMEL in each campaign for cross-validation. In total we compared

three POM instruments with the PFRs, two ESR network reference (master) instruments (one both in its initial and a later

modified version) and one travelling standard. In table 1 there is a summary with all instruments, the used datasets.

Table 1: The instruments used per location as reference and under study including the time periods of the common datasets. *
stands for a modified version of POMCNR that made it suitable for lunar observations

| Location | PFR Refer. instrument | Compar. Instrument | Starting date | End date |
|---|---|---|---|---|
| DAVOS I | N27 | POMVDV/CIMEL#354 | 09/08/2017 | 30/08/2017 |
| ROME I | N14 | POMVDV | 18/10/2017 | 02/11/2017 |
| ROME I | N14 | CIMEL646 | 05/12/2017 | 27/02/2018 |
| DAVOS II | N27 | POMCNR/CIMEL#354 | 24/07/2018 | 19/10/2018 |
| ROME II | N14 | POMCNR/POM11/CIME#L43 | 02/05/2019 | 03/10/2019 |
| DAVOS III | N27 | POMCNR*/CIMEL#916 | 08/10/2021 | 18/10/2021 |
| ROME III | N01 | POMCNR*/CIMEL#1270 | 03/09/2021 | 20/09/2021 |


### 2.1.1 PFR

The Precision Filter Radiometer (Wehrli, 2000) is a Sun photometer that measures the DSI in 4 wavelengths. The channels are

nominally centred on 368, 412, 501 and 862 nm. It is mounted on an independent tracking system to follow the motion of the

Sun. The instrument is covered with a quartz window and its internal parts are fully protected from the outside conditions. It



is filled with dry nitrogen at approximately 2 bar. Its temperature is kept constant at approximately 20ºC with an accuracy of 0.1ºC by an active Peltier system. The radiation passes through the quartz window and interference filters consequently to allow solar radiation from only a narrow spectral region reach the detector. The detector is a silicon photodiode that provides voltage measurements in mV proportional to the received light. Their full-width-at-half-maximum (FWHM) bandwidth varies from 3 nm to 5 nm and its field-of-view angle (FOV) is approximately 2º at FWHM. The four channels (filter – silicon

detectors) are arranged in a grid. Every minute a shutter opens for 10 seconds to perform the 10 sequential measurements at each wavelength, minimizing the exposure time of the filters to solar radiation hence their degradation. The stability of the travelling standard PFRs is validated by calibration of instrument before and after the campaigns.

### 2.1.2 PREDE-POM

The PREDE-POM (Estelles et al., 2012; Prede Co. Ltd., Japan: https://prede.com/english/skyradio.html) is a sun-sky

radiometer with a 2-axis stepping motor as tracking system to perform both direct sun and diffuse sky irradiance observations. The step is 0.0036º per pulse. There are 2 major versions of the instrument containing different wavelengths. POM-01 measures direct solar irradiance and diffuse sky irradiance at 7 wavelengths centred at 315, 400, 500, 675, 870, 940 and 1020 nm. POM-02 is an extended version measuring at 315, 340, 380, 400, 500, 675, 870, 940, 1020, 1627 and 2200 nm. In both cases the FWHM bandwidth is 2-10 nm depending on the channel. The wavelengths are isolated using filters mounted on a filter wheel

and the detector is a silicon photodiode except for the case of wavelengths above 1600 nm of POM-02, which are measured by a InGaAs photodiode. The FOV of the instrument is approximately 1º. It includes a temperature control system to keep the temperature at 30ºC, a 4-element silicon Sun sensor and a rain sensor. In this study, we used a standard POM-01 instrument and the rest were a modified POM-01 version to measure at 340 nm instead of 315 nm.

### 2.1.3 CIMEL

The CIMEL Sun-sky photometer (Giles et al., 2019) is an instrument including a 2-axis robotic tracking system. This tracking system allows it to perform direct sun and sky scans in order to measure either DSI and diffuse sky radiance. There are different versions measuring at different wavelengths. The smallest wavelength is 340 nm and the largest 1640 nm although for some versions it is 1020 nm. The number of wavelengths is up to 10. In this study we used CIMELs with at least 8 interference filters centred at 340, 380, 440, 500, 675, 870, 940, and 1020 nm. The bandwidth has full-width-at-half-maximum (FWHM)

of 10 nm, except for 340 and 380 nm which have 2 and 4 nm FWHM, respectively. To measure the radiation, it includes a silicon detector. The filters are mounted on a filter wheel that moves every second to switch to a different wavelength until all channels are measured in a measurement sequence. The measurement sequence is then repeated 3 times within 30 seconds to provide triplet observations. The instrument has a FOV of 1.2º. It has also a four-quadrant detector which detects the point of the maximum solar radiation intensity so it can point correctly to the Sun before the measurement sequence starts. The

AERONET AOD data are publicly available at 3 levels (1.0, 1.5 and 2.0). In this study, we used only level 2.0, which include cloud screening, the final calibration and quality assurance.



## 2.2 Calibration methods

We used 2 different calibration methods to calculate the extraterrestrial constant of the POMs. The Improved Langley Plot method (ILP) and calibration transfer using a PFR as reference.

### 2.2.1 Improve Langly Plot

The ILP method (Campanelli et al., 2004; Nakajima et al., 2020; Campanelli et al., 2023) is a modification of the conventionally used SLP. The basic principle in both methods is to use the solar radiation measured at the ground during at least half day and the Beer-Lambert-Bouguer law:

$$I = I_0 e^{-m\tau} \tag{1}$$

where $I$ the DSI measured at the ground, $I_0$ the calibration constant (solar irradiance at the top of the atmosphere in the units of the instrument) $m$ the air mass coefficient and $\tau$ the total optical depth of the atmosphere. The solar irradiance is measured at the instrument's units as the SLP and ILP methods do not require conversion to W/m$^2$. The total optical depth is the sum of the scattering and absorption optical depths of the atmospheric constituents.

Under no clouds in front of the solar disk:

$$\tau = \tau_R + \tau_g + \tau_a \tag{2}$$

where $\tau_R$ the Rayleigh scattering optical depth, $\tau_g$ the gas absorption optical depth and $\tau_a$ the extinction aerosol optical depth. Eq. (1) can be written as:

$$\ln I = \ln I_0 + m\tau \tag{3a}$$

or

$$\ln I = \ln I_0 - m\tau_R - m_g\tau_g - m_a\tau_a \tag{3b}$$

Knowing the atmospheric pressure, we can calculate $\tau_R$ and the total column of gases absorbing at a certain wavelength we can calculate $\tau_g$. $m_g$ and $m_a$ are the air masses corresponding to gases and aerosols.

The SLP uses Eq. (3a). By measuring the DSI during the day at several known air masses we can perform a linear fitting to the pairs of $m$ and $I$ values. The intercept of the fitted line is the natural logarithm of the calibration constant. This method

assumes that the total optical depth of the atmosphere is constant for at least several hours (slope of the linear fit), which does not happen in real conditions. At wavelengths where gas absorption is minor or the gases that absorb radiation show no rapid variability, AOD dominates the total optical depth. SLP at sun photometers (which use carefully selected wavelengths to avoid strong absorptions) is applicable with high accuracy in high altitude locations where the AOD is usually very low and its fluctuations do not have a significant effect on the total optical depth in timescales of a few hours. On the other hand, SLP

cannot be used in the aerosol polluted sites (Shaw et al., 1983; Toledano et al.,2018). In order to avoid the shipment of instruments to such locations and increase the frequency of calibration and monitoring its status, we require a method that is



usable at the station where the instrument is operated. ILP was developed for this purpose. Instead of using Eq. (3a) we can use a modified version of Eq. (3b).

Considering the Rayleigh scattering and gas absorption optical depths known, $\tau_a$. is the only required parameter to be retrieved before we calculate the calibration constant. In the ILP method instead of $\tau_a$ the used parameter is the scattering aerosol optical depth ($\tau_{sc}$). If $\omega$ is the single scattering albedo (SSA) then $\tau_{sc} = \omega\tau_a$, which leads to Eq. (3b) taking the form:

$$lnI + m\tau_R + m\tau_g = lnI_0 - m\frac{\tau_{sc}}{\omega} \tag{4}$$

Assuming $y = lnI + m\tau_R + m\tau_g$ and $x = m\tau_{sc}$ we get a straight line $y = ax + b$ where the slope is $a = -\frac{1}{\omega}$ and $b = lnI_0$.

Therefore, calculating $\tau_{sc}$ for several times during the day we can apply a linear fitting to all pairs of x and y values and calculate the calibration constant. This method takes into account the variability of the AOD but assumes constant SSA during the measurement period instead. Therefore, large variability of SSA can affect the accuracy of the method.

The estimation of $\tau_{sc}$ is possible through inversion modelling (by Skyrad pack code version 4.2 in our case) applied to the angular distribution of normalized sky radiance (NSR) (Eq. 5) observed in almucantar geometry at scattering angles up to 30°. The NSR is defined in Eq. 5:

$$R(\theta) = \frac{E(\theta)}{m\Omega I} \tag{5}$$

where $E$ is the measured diffuse sky irradiance, $\theta$ the scattering angle, $m$ the air mass, $\Omega$ the solid view angle (SVA) of the instrument and I the direct solar irradiance.

The model estimates the $\tau_{sc}$ and aerosol phase function by retrieving the size distribution with a-priori refractive index. To model the radiative transfer and retrieve $\tau_{sc}$ the surface albedo (SA), the total ozone column (TOC) and the surface pressure (P) are also required as inputs.

The Skyrad code derives also $\omega$ and therefore $\tau_a$, but it is not used in the ILP calibration. However, it is used for a screening criterion as all values corresponding to $\tau_a \geq 0.4$ are rejected before the final calibration.

### 2.2.2 Calibration transfer and AOD calculation

To evaluate ILP we calibrated the POMs using a PFR as a reference for each case. For measurements of DSI from co-located instruments at the same wavelength with $I_1$ being the DSI at the ground measured from PFR, $I_2$ the DSI measured from POM the same time, $I_{01}$ the PFR calibration constant and $I_{02}$ the POM calibration constant:

$$\frac{I_{1(\lambda,t)}}{I_{2(\lambda,t)}} = \frac{I_{01(\lambda)}}{I_{02(\lambda)}} \tag{6a}$$

The POM calibration constant is:

$$I_{02} = I_{01}\frac{I_2}{I_1} \tag{6b}$$



Therefore, we used the raw signal ratio of the instruments for measurements with a maximum of 30 sec time difference and the known calibration of the PFR to calculate the calibration for POM. The calibration constants and raw signals are in the units each instrument measures and corrected for the Earth-Sun distance differences by shifting everything to 1 A.U.

The signal ratios were cloud screened with the PFR AOD cloud screening algorithm (Kazadzis et al. 2018a) and filtered visually for outliers and days with erroneous measurements. Due to a diurnal variation of the signal ratios we used only data

between 9-13 UTC. We also excluded all days with fewer than 20 measurements in this day interval and calculated point to point calibration for the rest. We removed all point calibrations outside 2 standard deviations of the points during each day in a loop until 2 standard deviations fall below or equal to 0.5% of the daily median calibration. If the remaining points are below 20, the day is rejected. Finally, we examined the point calibrations and their corresponding AOD further to reject any remaining days with erroneous calibration. From the quality assured datasets, we calculated the POM daily median calibration and their

monthly average (since ESR calculates monthly the calibration with ILP).

To calculate the AOD we used the following procedure (used by ESR): We assumed that the monthly calibrations correspond to the last day of each month at 12:00 UTC. For measurements between 2 monthly calibrations, we use linear interpolation to calculate the calibration at the time of the measurement. For the first month of each campaign, we use the monthly calibration constant for all measurements of the month. We used only 2 wavelengths (500 and 870 nm) as they are directly comparable

between the instruments. The actual wavelength of each instrument may vary. The first channel has the same nominal value for both instruments (500 nm) and the difference of the actual central wavelengths may vary by less than 1-2 nm. For the second channel the nominal wavelength of the PFR is 862 nm, while for POM it is 870 nm. However, the Rayleigh and Mie scattering are weaker for longer wavelengths so the effect of approximately 8 nm difference is less significant at this spectral region.

**2.3 Intercomparison**

**2.3.1 Measurement intercomparison**

In order to assess the effect of calibration differences on AOD we compare the AOD of POMs retrieved from different calibrations at 500 nm and 870 nm. There are two AOD datasets for each POM: the original AOD provided by ESR and the AOD calculated from the calibration transfer. The two sets of monthly calibrations used and their differences are present in

the supplement table S1. These 2 AOD datasets differ also on AOD calculation algorithm (Kazadzis et al., 2018a). The ESR algorithm provides as AOD for a given moment the average of three consecutive measurements in one minute. In the calibration transferred based dataset we use the AOD from the raw signals corresponding to individual measurements. Also, the second dataset has no correction for the nitrogen dioxide ($NO_2$), while SKYNET takes $NO_2$ into account. Finally, there are differences regarding the pressure and ozone column values. We screened the data for clouds according to the GAW-PFR

algorithm. The reference AOD in all cases is the PFR AOD.





We added the co-located CIMEL instruments in the comparison as a third independent instrument taking advantage of the long-term experience on the observation of AOD differences between AERONET and GAW-PFR (Kazadzis et al., 2018a, Cuevas et al., 2019, Karanikolas et al., 2022). The CIMEL data were cloud screened by the AERONET algorithm and we further screened them according to GAW-PFR algorithm.

We use as indicators of the AOD differences the median difference, the standard deviation of the differences and their 5th and 95th percentile. According to the World Meteorological Organization (WMO) instruments are considered traceable when at least 95% of the AOD differences are within specific limits (WMO/GAW, 2005) given by the following Eq. 7:

$$\lim = \pm(0.005 \pm 0.01/m) \tag{7}$$

where $m$ is the air mass coefficient. Therefore, another indicator we used for the comparison is the percentage of data within
the WMO limits.

**2.4 Investigation on potential ILP error sources**

As the findings presented in Campanelli et al., 2023 and section 3.1 of the present study showed systematically negative difference between the ILP calibration and PFR based calibration transfers that are always larger in Rome, we investigate several potential causes.  Initially, we explore whether the aerosol properties between the two locations show any systematic
difference in terms of value and variability. We also assess the sensitivity of the ILP method to the pre-assigned values of six input parameters: solid view angle (SVA), surface pressure (P), total ozone column (TOC), surface albedo (SA), the real and the imaginary part of aerosol refractive index (RRI and IRI). Finally, we investigate whether the AOD, sc-AOD and SSA retrieved from the inversion modelling can provide evidence that may lead to explanation of the observed differences. In the sections below, we describe the methodology of the three aforementioned parts of the investigations.

**2.4.1 Aerosol properties**

There are three parameters which we included in this section. AOD, SSA and Angström Exponent (AE). According to Nakajima et al., 2020 the level of AOD affects the ILP performance. Also, the ILP method uses a pre-assigned refractive index value and assumes a stable SSA (which is connected with IRI) during the half day the ILP is performed (Eq. 4). Therefore, the SSA value and variability may affect the calibration. Due to the above, we assess whether there is an association of the levels
or the variability of AOD and SSA with the differences between ILP and the calibration transfer-based calibrations. For the AOD we used the PFR dataset. For the SSA the AERONET level 1.5 retrievals, due to lack of data availability of the quality assured level 2.0.  Because of the still limited SSA dataset and the larger uncertainty (compared to level 2.0) we also added the AE from the PFR in the investigation. AE is related to the size of aerosols. A change to AE reflects a change to aerosol composition that may affect IRI and SSA as well. For the AOD and AE we used only data corresponding to the half days used
for ILP calibrations. Additionally, we removed all points corresponding to AOD ≥0.4 at 500 nm and air masses ≥3, according to the screening criteria of the ILP method. For the SSA we used all data during the months of the campaigns except those corresponding to AOD at 440 nm <0.1 and a very small number of outliers. Since ESR provides monthly calibrations, we used



the monthly median values as indicator of the AOD, SSA and AE average levels. Each monthly median is the median of the daily medians. As indicators of the variability during the ILP method, we use the discrepancies between the monthly medians

of the daily 5th, 20th, 80th and 95th percentiles.

### 2.4.2 Sensitivity of ILP on input parameters

As the ILP calibration requires the instrument solid view angle (SVA), the surface pressure (P), the total ozone column (TOC), the surface albedo (SA), the real and the imaginary part of aerosol refractive index (RRI and IRI) as inputs, we examine to what extent they affect the ILP calibration.

The Skyrad 4.2 code use pre-selected values by the user for each of the last 5 parameters (P, TOC, SA, RRI and IRI). Surface pressure depending on the altitude of the station is provided by the Eq. 8:

$$P = P_0 e^{-0.0001184h} \qquad (8)$$

where P is the pressure in atm, $P_0$=1 atm and $h$ the altitude in meters. TOC is fixed to 300 DU for both Davos and Rome. SA is fixed to 0.1 (at non-polar regions like the ones in the present study), RRI 1.5 and IRI 0.005 for all wavelengths (340, 400,

500, 675, 870 and 1020 nm).

The SVA is derived with the disk scan method, an on-site calibration procedure (Nakajima et al., 2020; Campanelli et al., 2023). To investigate the effect of these input files we performed a set of ILP calibrations under different conditions in 3 sub-studies. For this section, we used only data from QUATRAM II as it is the longest campaign.

In the first sub-study we focus separately on each a-priori parameter of ILP calibration. We keep all other parameters in their

original values and change only the parameter under study. The goal is to recalculate the ILP calibrations for the local conditions of the station. Therefore, for each parameter under study we select a value based on observations in the measurement site. Specifically, TOC and P are present in the PFR data. TOC is taken OMI overpass (aura_omi_l2ovp_omto3_v8.5 https://acd-ext.gsfc.nasa.gov/anonftp/toms/omi/data/overpass/) and P was measured by a Setra barometer (uncertainty of less than 10 mbar). The refractive index parts (RRI and IRI) are available from the AERONET almucantar scans datasets only at

440, 675, 870 and 1020 nm. SA is also taken from the AERONET datasets in the same wavelengths and over land originates from a Li-Ross bidirectional reflectance distribution function (BRDF) model (Lucht & Roujean 2000) based on MODIS (or Moderate Resolution Imaging Spectroradiometer) satellite observations (Sun et al., 2017). For the rest of the wavelengths (340, 400, 500 and 940 nm) we had to select values based on the existing wavelengths either by interpolation and extrapolation (we used linear) (RRI, IRI) or by a separate criterion (SA). The SA selection is based on the observed SA and the spectral

dependence of the SA in the IGBP library from the LibRadtran package (Emde et al., 2016). The SVA is provided by ESR. For each parameter we used three different values to calculate three different ILP calibration constants. We calculated one ILP calibration using the median (RRI, IRI) or the mean (TOC, P and SA) value during all the months of the three QUATRAM campaigns. The other two calibrations correspond to the value one standard deviation above and below each average.  For the SVA, we used the values provided by ESR for the first ILP calibration. The other two values are based on the maximum





difference observed between ESR SVA and other SVA calibration methods for POMs presented in Campanelli et al., 2023. In
the supplement (sections 3-5 and tables S4-S6) we present all the values used for the six input parameters.

In the second sub-study, we alter the values of all parameters simultaneously except SVA (we used the value provided by
ESR). The goal is again to adapt the input parameters to the site conditions. We calculated the ILP calibration in two separate
cases:

a) Average case:  1 calibration per month using the monthly average values used in the first sub-study for all five parameters
under testing (RRI, IRI, P, TOC and SA).

b) 'Selected' case:  1 calibration per month. Here we selected one of the three values used in the first sub-study for the same
five parameters. The selected values are those of the three that lead to larger calibration constant. We picked only 1 month per
location for this case. The values of the input parameters used for this second sub-study are shown in the supplement section

315    6.

In the third sub-study, we tested the IRI, SA and SVA for a more extensive number of values (seven fixed values regardless
the location) to assess the behaviour of the calibration. For IRI and SA the selection includes is based on the three values of
the first sub-study, the 5th-95th percentiles of the observations and minimum/maximum values appeared. We also added semi-
arbitrary values between the observed and two extreme values (one very small and one very large) to test the performance of

the method at a wider range of inputs. For the SVA we use values based on the differences between the different SVA
calibration procedures appearing in Campanelli et al., 2023. The actual values for each parameter are in the supplement section
10 table S11.

**2.4.3 Investigation on the aerosol optical depth retrievals from sky radiance**

Since the ILP method is performed using linear fit of the logarithm of DSI with respect to the product of air mass coefficient

and scattering aerosol optical depth (sc-AOD) (Eq. 4), errors on the retrieval of the sc-AOD will transfer error to the calibration.
Since there is no reference dataset available for the sc-AOD, we tried to indirectly investigate potential errors using any
available data.

The Skyrad code retrieves both sc-AOD and SSA through inversion modelling and calculates the corresponding AOD as
additional information. Therefore, initially we compare that AOD dataset with the PFR AOD for potential differences.

However, systematic underestimation or overestimation on both sc-AOD and SSA retrieval can result in opposite errors to the
corresponding AOD that cancel each other. Due to the limitations of the AERONET SSA dataset (lack of level 2.0 data and
limited retrievals per day) we cannot evaluate the SSA retrieved by Skyrad 4.2 with confidence. Also, part of the SSA
difference between the AERONET product and the output of Skyrad code for ILP calibration may be attributed to the fixed
refractive index and the different scattering angles in the almucantar geometry used for the sky radiance measurements (ILP

uses only forward scattering having a maximum angle of 30 degrees).

Another indirect method to investigate the effect of the sc-AOD retrievals on the calibration performance is to use a different
inversion model to retrieve sc-AOD and re-calibrate the instrument with ILP. For this purpose, we used the inversion model





Skyrad pack MRI version 2 (Kudo et al, 2021). MRI allows the modelling of non-spherical particles in contrast to Skyrad pack 4.2 retrievals. It also introduced constraints at the edge of the size distribution to be stable and different smoothness constraints

(Kudo et al., 2021 provide detailed description). As mentioned in the same study, the MRI method is more accurate at high AOD. Under low AOD conditions in Davos a noticeable portion of data showed large retrieval error and unrealistic sc-AOD/AOD values. However, in both locations there were sufficient data to recalculate the ILP calibration and we applied it to the data of QUATRAM II.

We also investigated whether the variability of the SSA corresponding to the Skyrad 4.2 and MRI retrieval shows any

association with the calibration differences.

All retrieved AOD, sc-AOD and SSA data are screened according to the ILP criteria: keeping only data corresponding to AOD at 500 nm <0.4 and air mass <3.

## 3 Results

In this section we present the main findings of the study. First, we show the AOD differences between the CIMEL or POM

using different calibrations and the reference PFR. Then we present the stability and uncertainties of the used calibrations. Finally, we attempt to investigate the causes of the observed differences through the methodology described in section 2.4.

### 3.1.1 AOD intercomparison

There are three campaigns per location and we present the AOD differences between the PFR and POMs or CIMEL. In Fig. 1 we show the median AOD differences and standard deviation (box size) and the 5th and 95th percentiles of the differences

(error bars). Something evident is that the ESR AOD calculated with ILP is systematically lower than the PFR AOD. In Davos the median differences are between -0.06 and -0.01 at 500 nm and 0.000 to -0.005 at 870 nm. In Rome the median differences range from approximately -0.014 to -0.034 at 500 nm with the vast majority of differences below -0.01. At 870 nm QUATRAM I in Rome shows a median difference of -0.005 and the other cases below -0.01. For QUATRAM II in Rome which was the longest campaign and the one with the largest differences at the POM master (POMCNR), we included a second POM

(POM11), which shows performance similar to the POM master (POMCNR*) of QUATRAM III in Rome.

When using calibration transfer from PFR to recalculate the AOD for POMs the absolute median differences are below 0.005 for all cases. Most of the times in the case of calibration transfer the median difference remains negative, but there are exceptions. The CIMEL-PFR comparison shows similar results with all median AOD differences below 0.01. Also, the majority of 5th-95th percentiles for either CIMEL-PFR or POM-PFR using calibration transfer are within 0.01.

Regarding the WMO traceability criteria, the data within WMO limits for POM AOD with ILP calibration are below 95% for all cases at 500 nm and QUATRAM II and III in Rome at 870 nm (table 2). However, there is large deviation between the two locations where at 500 nm the percentage in Davos is above 60%, while in Rome below 4%. Using the calibration transfer to calculate POM AOD, in all cases there are more than 98% of data within the WMO limits (table 2). The CIMEL-PFR





comparison (table 3) also shows percentage mainly above 98%. Exceptions are the QUATRAM I and II phases in Rome at
500 nm and QUATRAM I Rome phase at 870 nm. All cases have at least ~60% of differences within the limits.

Recalculating the AOD with the same post processing algorithm and for the same instrument (once for each POM) for the two POM calibrations (ILP and calibration transfer) we can observe purely the effect of the calibration on AOD. In that case, the median AOD difference in that case is similar to the difference between the original POM and PFR datasets shown in Fig. 1 green boxes. The results of the comparison showing the calibration effect along with the 'original' differences are in the
supplement in Fig. S1 (section 1). The median AOD differences attributed to calibration deviate from the 'original' AOD differences by less than 0.003 aside from three cases. It is approximately 0.005 for QUATRAM III Rome phase at 500 nm and Davos phase at 870 nm. It is almost 0.01 for QUATRAM II Rome phase at 500 nm of only one of the POMs (POM_CNR). These deviations are not systematically larger or smaller than the 'original' at 870 nm, but they are smaller for most campaigns at 500 nm.

The variability of AOD differences in the case of the comparison between the two recalculated POM AOD datasets (which show purely the calibration effect), is a result of the dependence of the calibration effect on the air mass. Therefore, it depends on the magnitude of the calibration difference, its month-to-month variability and the air mass distribution present on the data. These results suggest that the post processing algorithm and instrument technical differences between the networks are a source of only random AOD differences within the retrieval uncertainty. In the case of ESR the calibration method difference
dominates the overall AOD difference.

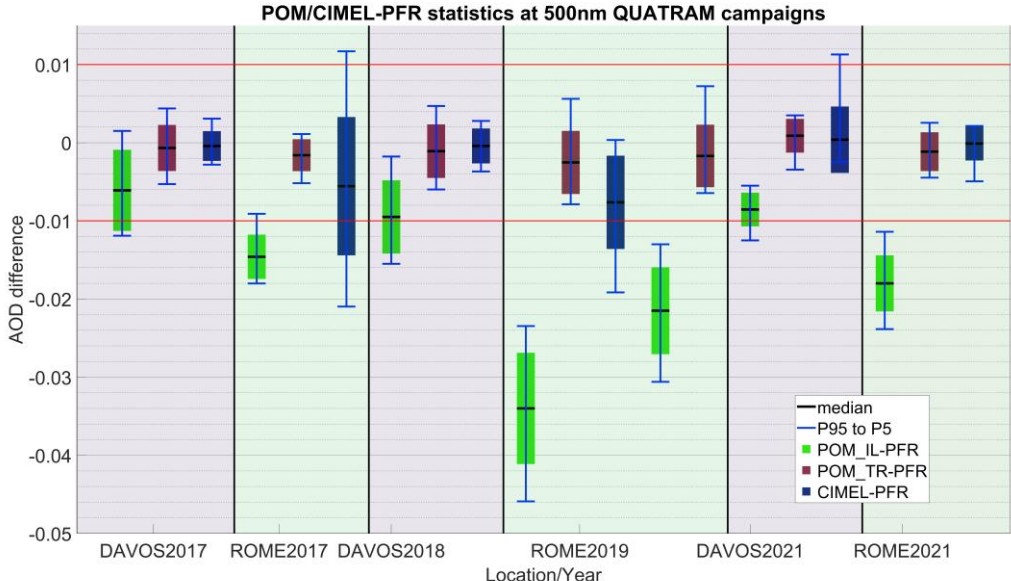

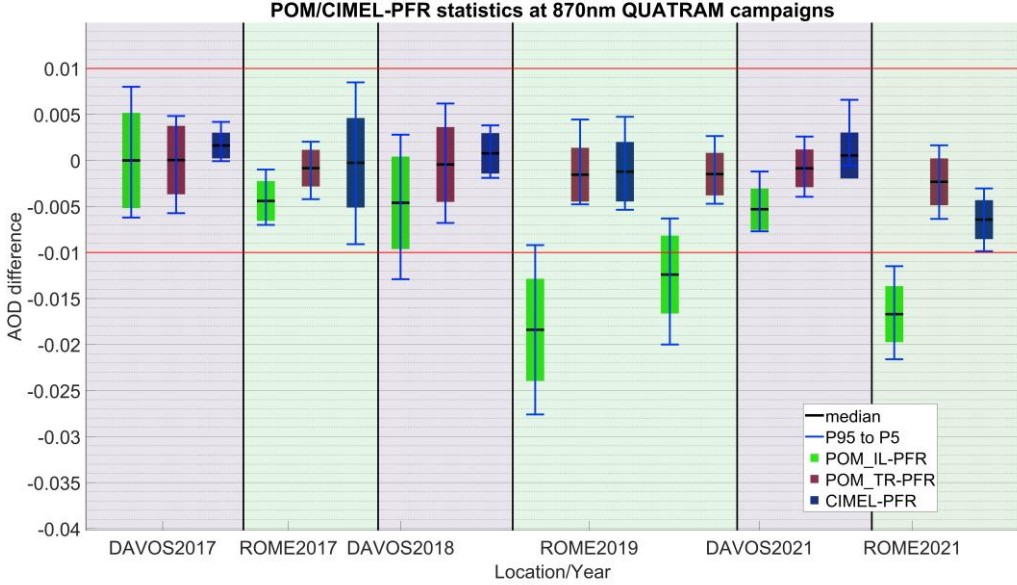

**Figure 1: Box plot of the AOD differences' statistics for all instrument comparisons during both phases of the 3 QUATRAM campaigns. The black line is the median difference, the size of the boxes denotes the distance between the median and the standard deviation, while the error bars show the 5th and the 95th percentile of the AOD differences. In all cases the PFR AOD is the reference instrument. The green boxes correspond to the differences between the original AOD from POMs and the reference. The red boxes correspond the POM AOD calculated with the calibration retrieved with transfer from the PFR. The blue boxes correspond to the differences between CIMEL and PFR. For the Rome 2019 campaign we compare 2 different POMs with the same PFR (left POM_CNR and right POM11). Top: 500 nm. Bottom: 870 nm.**



Table 2: The percentage of AOD differences within WMO limits for the comparison between PFRs and POMs. IL refers to the original POM AOD retrieved using the ILP calibration method and TR to the calibration transfer based AOD.

| Location | Instrument | Year | Number of measurements | WMO limits % IL | | WMO limits % TR | |
|---|---|---|---|---|---|---|---|
| | | | | 500 nm | 870 nm | 500 nm | 870 nm |
| DAVOS I | POMVDV | 2017 | 1929 | 84.34 | 95.23 | 99.74 | 98.65 |
| DAVOS II | POMCNR | 2018 | 6604 | 63.51 | 89.13 | 99.03 | 98.21 |
| DAVOS III | POMCNR* | 2021 | 1516 | 72.1 | 99.47 | 100.00 | 100.00 |
| ROME I | POMVDV | 2017 | 507 | 3.16 | 99.01 | 98.62 | 100.00 |
| ROME II | POMCNR | 2019 | 3903 | 0.00 | 11.48 | 99.95 | 99.95 |
| ROME II | POM11 | 2019 | 6079 | 2.66 | 44.56 | 99.10 | 100.00 |
| ROME III | POMCNR* | 2021 | 904/908 | 2.99 | 1.32 | 100.00 | 100.00 |

Table 3: The percentage of AOD differences within WMO limits for the comparison between PFRs and CIMELs.

| Location | Instrument | Year | Number of measurements | WMO limits % | |
|---|---|---|---|---|---|
| | | | | 500 nm | 870 nm |
| DAVOS I | CIMEL#354 | 2017 | 614 | 99.84 | 99.84 |
| DAVOS II | CIMEL#354 | 2018 | 1127 | 99.38 | 99.47 |
| DAVOS III | CIMEL#916 | 2021 | 271 | 100.00 | 100.00 |
| ROME I | CIMEL#646 | 2017/2018 | 117 | 59.83 | 90.60 |
| ROME II | CIMEL#43 | 2019 | 2278 | 75.20 | 100.00 |
| ROME III | CIMEL#1270 | 2021 | 243/253 | 100.00 | 98.81 |

### 3.1.2 Calibration stability and uncertainties

In the previous section we show that the major source of AOD differences between the PFRs and POMs is the calibration method difference. The calibration differences between the ILP method and the PFR-based transfer can be found in the supplement table S1 (section 1). The values in the supplement show some minor differences compared to Campanelli et al., 2023 for some months mainly due differences in the day selection that are larger for August 2018 in Davos (where we observed an abrupt calibration shift during the month and removed the days before the shift as the monthly calibration is attributed to





the end of the month when retrieving AOD). In this section, we present the stability and the uncertainties of the different
calibrations.

The ILP calibrations show either positive and negative fluctuations for consecutive months in the same location between 0.17-
2.3% with a median absolute value of 0.55% and a standard deviation of 0.87%. It can be attributed both to changes in the
instruments and the random uncertainty of the ILP method. An estimation of the uncertainty magnitude is evident in the
coefficient of variation (CV%) of the daily ILP calibrations per month (Campanelli et al., 2023 preprint table 2a) which are
between 0.18%-2.87% at 500 and 870 nm.

The PFR calibration differences between consecutive calibrations are between 0.00-0.45% at 500 and 870 nm (supplement
table S3). All calibrations have uncertainty below 0.4% (supplement table S2).

The PFR based calibration transfers of POMs show fluctuations for consecutive months in the same locations between 0.00-
1.72% with a median absolute value of 0.19% and a standard deviation of 0.56%. The uncertainties of the calibration transfers
as the combination of the PFR calibration uncertainty $\sigma_{PFR}$ and the standard deviation of the daily calibrations $\sigma_d$ are calculated
as:

$$\sigma_{TR} = \sqrt{\sigma_{PFR}^2 + \sigma_d^2} \qquad\qquad (9)$$

The calibration transfer uncertainties are between 0.27%-0.8% (supplement table S2).

The fluctuations of ILP and transfer-based calibrations do not coincide, which is reflected in the month-to-month fluctuations
of their difference being 0.01%-1.93% with median absolute value of 0.55% and standard deviation 0.96%.

However, not all fluctuations can be explained by the presented uncertainties. A particularly interesting case is the calibration
change from July to August 2019 in Rome for POMCNR at 870 nm. The CV% of the ILP calibrations of these two months is
below 0.5% (Campanelli et al., 2023), while their calibration difference is 1.3%. The calibration transfers from the PFR for
the same months differ only by 0.2% providing no evidence of changes in the instrument. The same months show an ILP
calibration change above 2% for POM11, with the calibration transfers differing by 0.3%. At 500 nm for the same months the
ILP differences above 1%, while the calibration transfer differences are 0%. Therefore, the ILP differences between these two
months are attributable to the overall uncertainty of ILP.

## 3.2 Investigation on calibration differences

As shown in section 3.1.1 the ESR dataset shows a systematic AOD underestimation compared to GAW-PFR and AERONET
due to an underestimation in the calibration from the ILP method. However, this calibration difference varies significantly





between the two locations and from month to month. Using the methods described in section 2.4 we attempted to explain why this underestimation happens and why it is systematically larger for Rome.

### 3.2.1 Aerosol properties

Here we investigate whether there is any systematic difference between Davos and Rome on AOD, SSA and AE values or 440 variability that could potentially be associated with the larger calibration differences in Rome for all months. We use AOD and AE from the PFR data during the half/full days of the ILP calibrations and SSA is from the AERONET data during the QUATRAM campaigns. We used monthly medians as the average level and monthly medians of the daily percentiles ($5^{th}$, $20^{th}$, $80^{th}$ and $95^{th}$) as variability indicator as described in section 2.4.1.

### 3.2.1.1 Aerosol Optical Depth

Here we present the PFR AOD values for all months of the campaigns in both locations. The results are visible in Fig. 2, where the green boxes correspond to 500 nm and the red to 862 nm. For most months it is evident that the AOD is higher and more variable in Rome, but there are exceptions like QUATRAM I (DAV17/ROM17). Also, we can see that the highest AOD corresponds to QUATRAM III in Rome (ROM21) while the largest calibration and AOD differences between PFR and POM were in QUATRAM II (ROM19). Both AOD values and variability vary within the same location and between the two from 450 month to month showing no consistency between AOD (Fig. 1) and calibration differences (supplement, table S1).

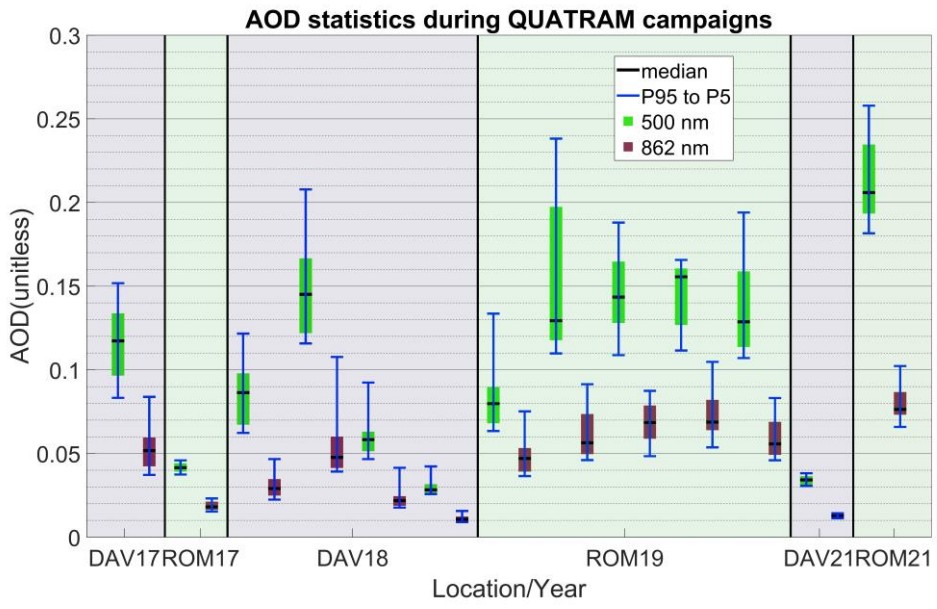





**Figure 2: The AOD statistics for all months of all campaigns. The green boxes correspond to 500 nm and the red to 862 nm. The extent of the box shows the median of the 20th and 80th percentiles per day and the error bars the median of the 5th and 95th percentiles per day. Each box is 1 month of the campaign.**

### 3.2.1.2 Single Scattering Albedo

ILP assumes a constant SSA as the inverse slope the linear fit (section 2.2.1) and the refractive index pre-assigned to specific value which potentially reduces the accuracy of the method. Here we present the AERONET SSA values and variability between the months of the campaigns (Fig. 3) at 440 nm (green) and 870 nm (red). For Davos 2018 campaign there are three months instead of four as there was lack of data during the first month (July 2018) since the campaign started towards the end of the month. Generally, no systematic difference between the two locations is evident nor an association between the

calibration and AOD differences even for the same location. In Rome the largest SSA variability corresponds to QUATRAM I (ROM17) in which we observed the smallest calibration and AOD differences during the Rome phases. Similarly, in Davos the largest variability in during QUATRAM III (DAV21), which also exceeds the Rome SSA variability. However, we did not observe larger differences between ILP and calibration transfer in Davos during QUATRAM III (DAV21) compared to QUATRAM II (DAV18). In terms of median SSA, depending on the month, either Rome or Davos may have larger SSA. The

fluctuations of SSA do not seem to significantly affect the calibration differences. However, we acknowledge that the limitations of the SSA (2.4.1) limit the confidence of conclusions.



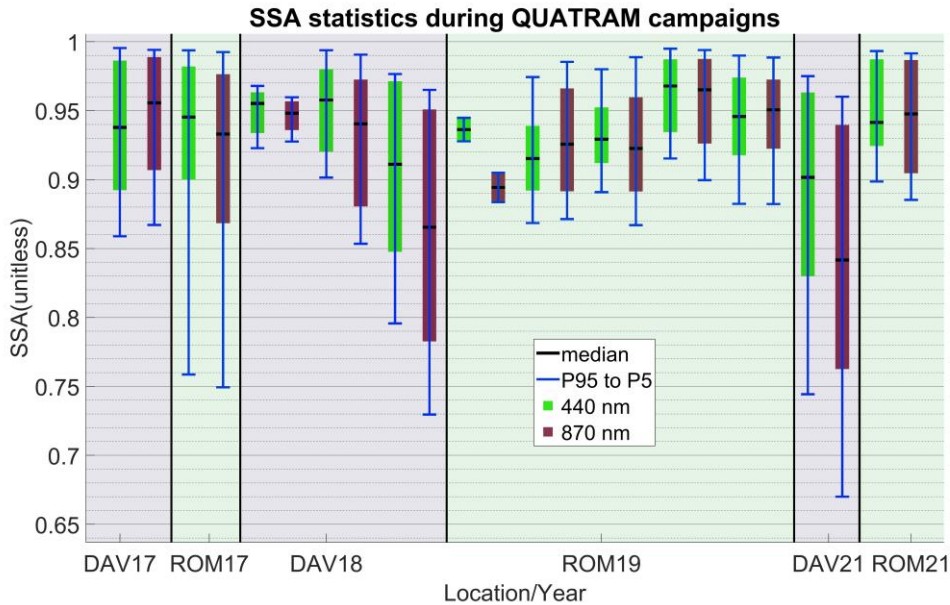

**Figure 3: The SSA statistics for all months of all campaigns. The green boxes correspond to 440 and the red to 870 nm. The extent of the box shows the median of the $20^{th}$ and $80^{th}$ percentiles per day and the error bars the median of the $5^{th}$ and $95^{th}$ percentiles per day. Each box is 1 month of the campaign. In QUATRAM II (DAV18) the first month of the campaign (July) is missing due to lack of data.**

### 3.2.1.3 Angström Exponent

Due to the limitations of the SSA dataset (section 2.4.1), we added a comparison of the AE medians and variability during the campaigns as an indicator of aerosol composition. The results are in Fig. 4 with green corresponding to Davos and red to Rome. During QUATRAM I (DAV17/ROM17) the two locations have very similar median AE, but Davos shows the largest variability. During QUATRAM II (DAV18/ROM19) the AE in Davos is the largest, while the variability varies significantly between the months. Similarly, during QUATRAM II in Rome AE is lower and each variability largely depending on the month. Finally, during QUATRAM III (DAV21/ROM21) Rome shows the largest AE and variability. Again, there is no systematic difference between the two locations nor an association of calibration differences and AE within the same location.





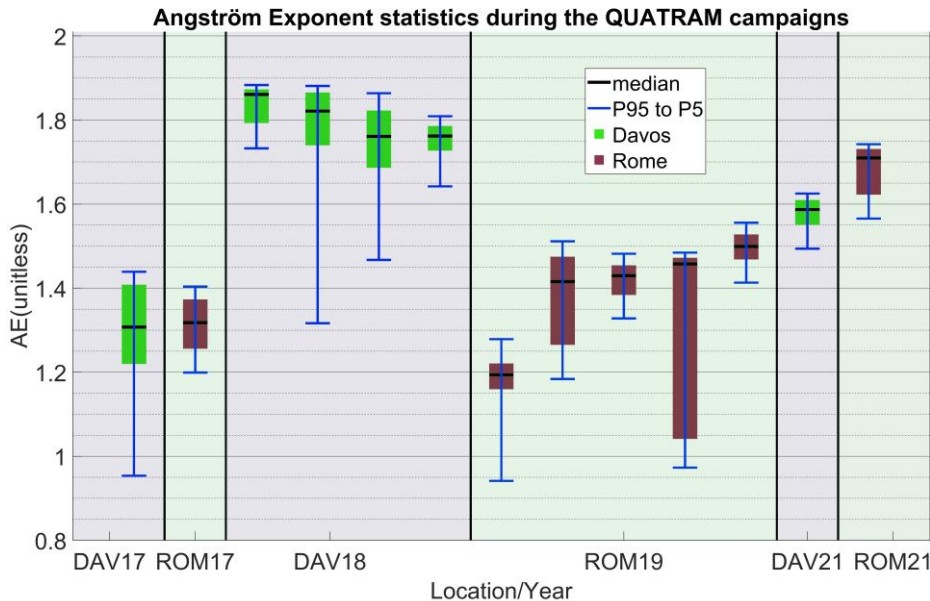

**Figure 4: The AE statistics for all months of all campaigns. The green boxes correspond to Davos and the red to Rome. The extent of the box shows the median of the 20th and 80th percentiles per day and the error bars the median of the 5th and 95th percentiles per day. Each box is 1 month of the campaign.**

### 3.2.2 Sensitivity of ILP on input parameters

As the available aerosol conditions during the campaigns show no indication of an explanation to the ILP underestimation and the differences between locations, we attempted to investigate the causes through a sensitivity study of the ILP. ILP uses six parameters as inputs: Real part of refractive index (RRI), Imaginary part of refractive index (IRI), Surface albedo (SA), Total Ozone Colum (TOC), Surface Pressure (P) and Solid View Angle (SVA). The first five are pre-selected and the last is provided by an in-situ calibration method. Therefore, there are discrepancies between the real atmospheric conditions under which the ILP is performed and the selected values.

#### 3.2.2.1 ILP Test based on local observations: one variable parameter per case

Here we present results of the ILP calibration using different values for the input parameters of Skyrad 4.2. The selection is described in section 2.4.2.

The RRI average observations from AERONET were very close to the pre-assigned input of Skyrad pack 4.2 (1.5 for all wavelengths) and the standard deviation small, so we used the average, minimum and maximum values (1.33 and 1.6). The calibration difference due to this change in the RRI were between 0.00-0.21%.





For the surface pressure (P) we used the values 0.8, 0.83 and 0.85 atm for Davos, while 0.97, 1 and 1.02 atm for Rome (the middle value is the one used originally for ILP). Most differences were below 0.05%. During one month at 870 nm, we obtained

the maximum difference of 0.2% (July 2019 in Rome where the maximum sensitivity at RRI was also present).

For total ozone column (TOC) we used both locations 260, 300 and 400 DU, which resulted again in differences were below 0.05% except July 2019 at 870 nm.

Due to the small sensitivity at these three parameters, we do not include a more detailed analysis on them, but the comparisons are available in the supplement (sections and tables S8-S10). For the imaginary part of refractive index (IRI), surface albedo

(SA) and solid view angle we observed cases of larger sensitivity.

In the Fig. 5-7 we can see the calibration differences between ILP runs and the calibration transfer from PFR for different conditions. The results correspond to the first sub-study described in section 2.4.2 where we study each parameter separately according to the observations of each site. The results correspond to all months of QUATRAM II.

For the majority of the cases the calibration differences due to IRI are smaller than 0.5% (Fig. 5). For specific months (August

2018-Davos and July 2019-Rome) it is 1% or higher. However, a calibration difference between ILP and calibration transfer of 1% in Davos and 2.5% in Rome at 500 nm and above 1.5% in Rome at 870 nm remains even for those particular months.

Using the SA from AERONET reduces the calibration difference noticeably (Fig. 6) at 500 nm for most months in both locations, but the effect can explain a calibration difference of approximately up to 0.75% (September 2019, Rome), while the calibration differences in Rome are between 2.5-3.5% (table S1 supplement).

In the case of SVA there are also noticeable differences of 0.5-1% from the central value (Fig. 7). SVA like IRI shows also particularly high sensitivity during the second month (August 2018, Davos). The central SVA value corresponds to identical all input parameters as the original calibration and therefore we expect the magenta line (original) in fig. 7 and the blue (central SVA) to be identical. Some differences below 0.1% are present probably in most months due to the usage of different compilers and versions of the Skyrad pack 4.2. However, for September 2019 in Rome at 500 nm they differ up to 0.5% and August

2018 in Davos at 870 nm above 1%. This may be a result of computational instability or modifications in the Skyrad pack 4.2 screening criteria for the selection of data to perform the ILP since the time the instruments were initially calibrated. For the rest of the months such differences are below 0.1%.






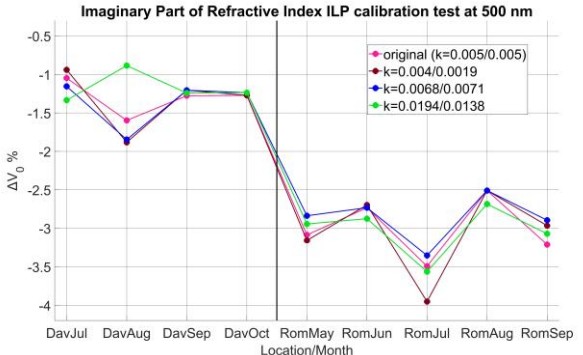 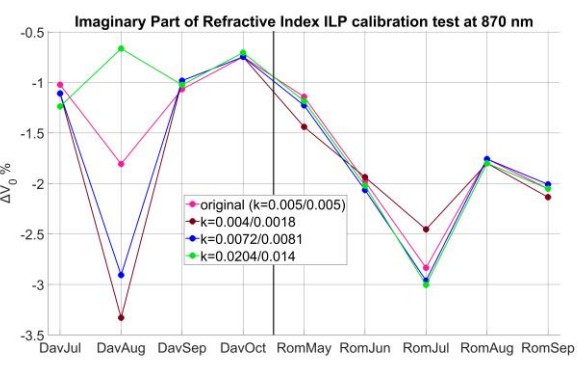

**Figure 5:** The %difference between IL calibration and calibration transfer for POMCNR during the QUATRAM II months using different values of imaginary refractive index (original calibration, median k and median±std). Left 500 nm and right 870 nm. The left side from the black line corresponds to the Davos calibrations and the right side to Rome. The black line separated the Davos and Rome months (July to October 2018 and May to September 2019).


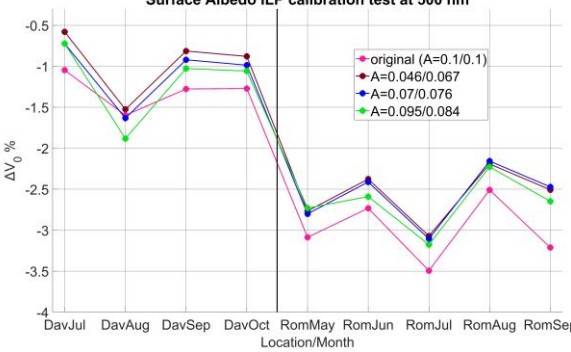 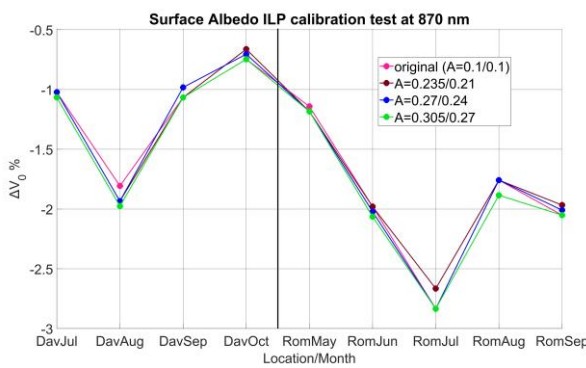

**Figure 6:** The %difference between IL calibration and calibration transfer for POMCNR during the QUATRAM II months using different values of surface albedo (original calibration, median A and median±std). Left 500 nm and right 870 nm. The left side from the black line corresponds to the Davos calibrations and the right side to Rome. The black line separated the Davos and Rome
months (July to October 2018 and May to September 2019).

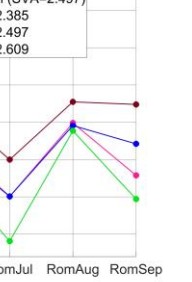 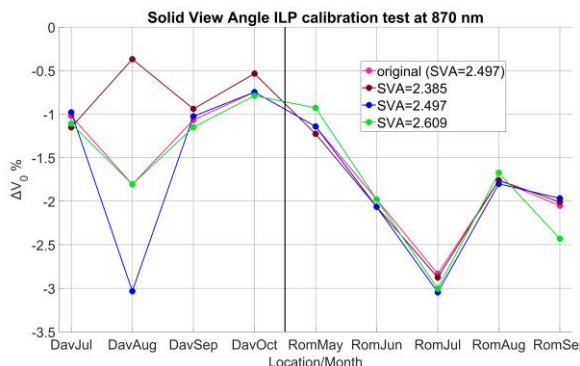





**Figure 7: The %difference between IL calibration and calibration transfer for POMCNR during the QUATRAM II months using**
**different values of solid view angle (original calibration, runs with the provided SVA and SVA± fixed deviation). Left 500 nm and**
**right 870 nm. The left side from the black line corresponds to the Davos calibrations and the right side to Rome. The black line**
**separated the Davos and Rome months (July to October 2018 and May to September 2019).**

### 3.2.2.2 ILP Test based on local observations: all parameters as variables

In the section we present the results of the second sub-study described in section 2.4.2 Here there are two cases of calibrations
that we tested in the whole QUATRAM II campaign.

The results on the table 4 show for the average case less than 0.5% changes with exception in the case of August 2018 in
Davos, due to the large sensitivity in the IRI the calibration changed more than 1%.

Under the 'selected' case (selected conditions for all parameters that increase the ILP calibration), there is larger increase of
the calibration in Davos and compared to Rome at both wavelengths (table 4). All differences are below 1%.

Table 4: The %difference between the original ILP and transferred calibrations minus the %difference between the ILP under
selected conditions and the transferred. Positive values indicate smaller difference between ILP and calibration transfers
compared to the differences of the original calibrations.

| Average case | | | | | |
|---|---|---|---|---|---|
| **Instrument** | **Location** | **Year** | **Month** | **$\Delta V_0$ % 500 nm** | **$\Delta V_0$ % 870 nm** |
| POMCNR | DAVOS | 2018 | 7 | 0.25 | -0.09 |
| POMCNR | DAVOS | 2018 | 8 | 0.14 | -1.27 |
| POMCNR | DAVOS | 2018 | 9 | 0.36 | 0.08 |
| POMCNR | DAVOS | 2018 | 10 | 0.29 | 0.08 |
| POMCNR | ROME | 2019 | 5 | 0.46 | -0.09 |
| POMCNR | ROME | 2019 | 6 | 0.36 | -0.26 |
| POMCNR | ROME | 2019 | 7 | -0.14 | -0.13 |
| POMCNR | ROME | 2019 | 8 | 0.32 | -0.04 |
| POMCNR | ROME | 2019 | 9 | 0.46 | 0.00 |
| **'Selected' case** | | | | | |
| POMCNR | DAVOS | 2018 | 9 | 0.89 | 0.34 |
| POMCNR | ROME | 2019 | 8 | 0.60 | 0.13 |




### 3.2.2.3 ILP sensitivity tests

In this section we present the results of the third sub-study described in section 2.4.2, where we only test IRI, SA and SVA for seven values over a larger range. We selected only one month per location avoiding the August 2018 and July 2017 due to the behaviour presented in the previous two sections. In the figures 8-10 are the results per parameter.

Changing only the IRI shows that the ILP changes less than 0.25% for both wavelengths and locations (Fig. 8) and IRI below 0.05. Increasing IRI to larger or even either rare or unrealistic values has no effect on the calibration. Therefore, ILP appears to be either affect it significantly or very little depending on the month.

Changing only the SA shows (Fig. 9) a monotonic, but non-linear dependence of the ILP calibration where larger SA leads to smaller calibration constant. At 870 nm there is a maximum calibration constant at SA 0.04 with approximately 0.07-0.08

being the average values from AERONET and 0.1 the values used by ESR. At 500 nm the difference between ILP calibrations in Davos and Rome also are reducing at lower SA showing that ILP in Rome is affected to a larger extent by the SA value at 500 nm. However, even when using a SA as low as 0.02 the remaining calibration difference between the calibration transfer and ILP at 500 nm is approximately 2% in Rome and 0.7% for Davos. At 870 nm the difference remains at least for 0.95% Davos and 1.7% for Rome.

Finally, in the case of SVA (Fig. 10) there is a monotonic decreasing dependency of the calibration constant and SVA, at 500 nm, while some fluctuations at 870 nm. The minimum calibration difference at 500 nm is approximately 0.58% for Davos and 1.7% for Rome, while at 870 nm 0.78% for Davos and 1.6% for Rome.

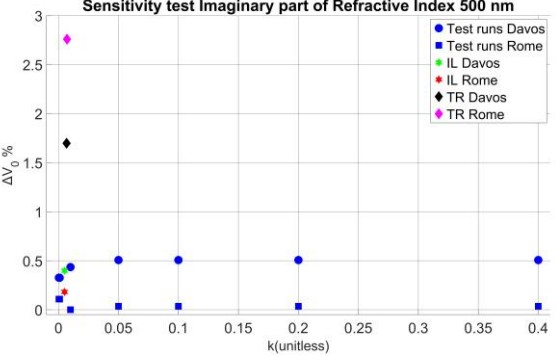
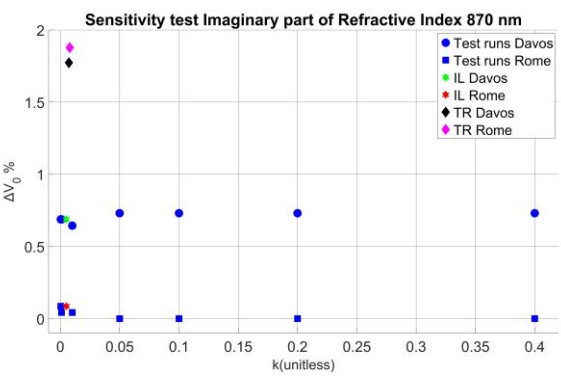


**Figure 8: Sensitivity test of the IL calibration on the imaginary refractive index at 500 (left) nm and 870 nm (right). The vertical axis shows the % difference of each calibration from the selected zero. As 0 we selected the lowest calibration constant of the sensitivity tests present in each graph. The blue squares correspond to Rome sensitivity runs, the blue circles to Davos, the stars to the original ILP calibration and the diamonds to the calibration constants from transfer with a PFR as reference.**


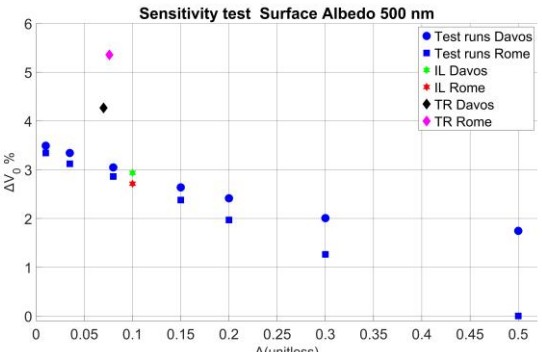
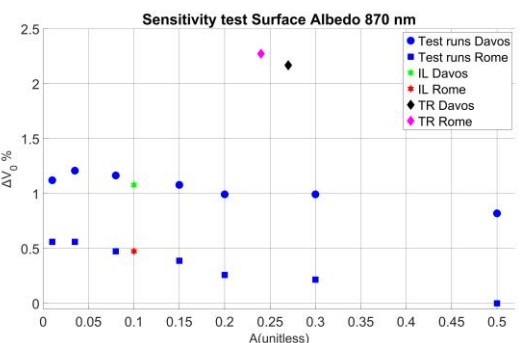

**Figure 9: Sensitivity test of the IL calibration on the imaginary refractive index at 500 (left) nm and 870 nm (right). The vertical axis shows the % difference of each calibration from the selected zero. As 0 we selected the lowest calibration constant of the sensitivity tests present in each graph. The blue squares correspond to Rome sensitivity runs, the blue circles to Davos, the stars to the original ILP calibration and the diamonds to the calibration constants from transfer with a PFR as reference.**

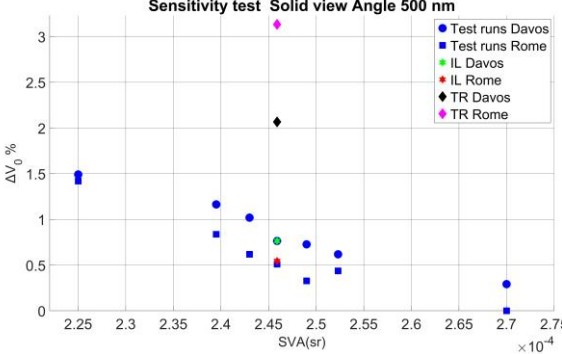
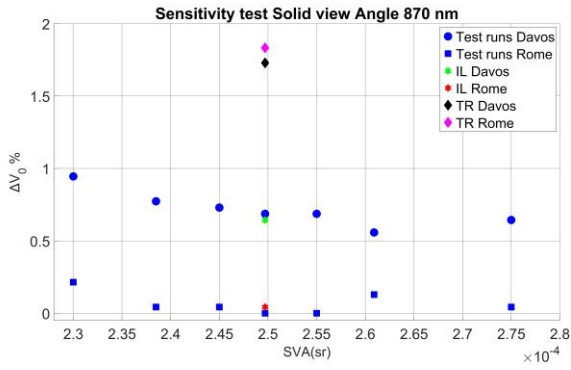

**Figure 10: Sensitivity test of the IL calibration on the imaginary refractive index at 500 (left) nm and 870 nm (right). The vertical axis shows the % difference of each calibration from the selected zero. As 0 we selected the lowest calibration constant of the sensitivity tests present in each graph. The blue squares correspond to Rome sensitivity runs, the blue circles to Davos, the stars to the original ILP calibration and the diamonds to the calibration constants from transfer with a PFR as reference.**

### 3.2.3 Investigation on the aerosol optical depth retrievals from sky radiance

As discussed in section 3.1.2 the ILP method can have significant random uncertainty as individual ILPs for half day leading to different values that are averaged monthly. However, the vast majority of daily calibration constants are lower than the calibration transfers from PFR and most of them by more than 0.5-1% (table 5) for both locations and wavelengths. This shows the significance of the systematic bias. One way to get such biased result, is a systematic underestimation in the sc-AOD provided by the inversion of NSR or an underestimation of sc-AOD in the small air masses and overestimation in the large air masses.




Table 5: The percentage of daily ILP calibration constants below the corresponding monthly calibration transfer (column 4), below the calibration transfer at least 0.5% (column 5) and at least 1% (column 6). The rows correspond to the days used for the final ILP monthly calibrations for each location under all campaigns at 1 wavelength.

| Wavelength (nm) | Location | Number of days | $\%\Delta V_0<0$ | $\%\Delta V_0<=-0.5\%$ | $\%\Delta V_0<=-1\%$ |
|---|---|---|---|---|---|
| 500 | DAVOS | 45 | 95.56 | 91.11 | 73.33 |
| 500 | ROME | 112 | 100.00 | 100.00 | 98.02 |
| 870 | DAVOS | 38 | 94.74 | 86.84 | 52.63 |
| 870 | ROME | 101 | 97.03 | 96.04 | 93.07 |

In this section we investigate the effect of sc-AOD retrieval through inversion of ILP performance. As there were two inversion algorithms available, we compare the calibration and the sc-AOD calculated by Skyrad pack 4.2 with calibration and sc-AOD from Skyrad MRI.

The AOD from Skyrad 4.2 is retrieved through the inverted sc-AOD it may show similar errors. Since we do not have a sc-AOD reference dataset we compared the Skyrad AOD with the PFR AOD.

The AOD difference between the AOD retrieved from the Skyrad pack 4.2 using almucantar scans of POM and the PFRs show a systematic underestimation as expected aside from the comparison at 870 nm for Davos (table 6). The differences are also higher in Rome. However, the median differences are significantly smaller than the ones corresponding to the ESR direct sun AOD product compared to the same PFRs and the percentage of differences within the WMO limits higher. The AOD differences are also increasing for smaller air masses in Rome, but not in Davos. For air masses below 1.5 the median AOD difference is -0.012/-004 at 500/870 nm in Rome and 0.000/0.001 at 500/870 nm in Davos. For air masses above 2, the median AOD difference is -0.005/-0.000 at 500/870 nm in Rome and -0.003 /0.000 at 500/870 nm in Davos. More details including linear fitting of the air mass dependencies are available in the supplement section 12 table S15.

Table 6: The statistics of the differences between the AOD from Skyrad pack 4.2 using POM almucantar scans and the AOD from PFR. The results correspond to all QUATRAM campaigns for each location. The time difference threshold is 30 seconds. Starting from the third column we show the median of all AOD differences, the percentage of differences within the WMO limits, the 5th and the 95th percentiles of AOD differences and the total number of measurements compared per location.

| Location | wavelength | median difference | WMO limits % | P5th | P95th | Number of measurements |
|---|---|---|---|---|---|---|
| DAVOS | 500 | -0.002 | 82.91 | -0.014 | 0.015 | 1129 |
| DAVOS | 870 | 0.000 | 97.25 | -0.004 | 0.007 | 1129 |





| ROME | 500 | -0.009 | 64.09 | -0.027 | 0.007 | 1231 |
| ROME | 870 | -0.003 | 92.85 | -0.012 | 0.009 | 1231 |

Using the sc-AOD from MRI as input to the ILP instead of the Skyrad 4.2 in Davos 2018 and Rome 2019 we obtained different calibration constants for each month, but there is no consistent improvement (table 7). At 500 nm 6 out of 9 months show calibration closer to the calibration transfers by between 0.29-0.96% (negative differences), while at 870 nm the calibration constant is increased only for 3 months (0.04-1.39%). The sc-AOD median differences are negative at 500 nm and positive at 870 nm, which is in accordance with the sign of the calibration differences for most cases. However, they are very small (up 640 to 0.002) and there is no consistency between sc-AOD and calibration differences (table 6). Due to the fact that the datasets are different, there is also different selection of individual sc-AOD inversions and days passing the criteria for the final ILP calibration. The combination of using randomly different sc-AOD, points and half day selections results to the calibration differences observed that are mainly below 1%. Such random differences are similar to the magnitude of ILP CV% in Campanelli et al., 2023.


Table 7: The % difference between the original ILP calibration and the ILP calibration using sc-AOD inverted by Skyrad MRI (columns 3 and 4) and the median differences of the corresponding sc-AOD (columns 6-7).

| Year | Month | $\Delta V_0$ % 500 nm | $\Delta V_0$ % 870 nm | Median $\Delta$sc-AOD 500 nm | Median $\Delta$sc-AOD 870 nm | Number of sc-AOD measurements |
| --- | --- | --- | --- | --- | --- | --- |
| 2018 | 7 | 0.40 | 0.17 | -0.002 | 0.000 | 194 |
| 2018 | 8 | -0.54 | 2.16 | -0.002 | 0.001 | 404 |
| 2018 | 9 | -0.96 | -0.64 | -0.002 | 0.000 | 332 |
| 2018 | 10 | -0.54 | -1.39 | -0.002 | 0.000 | 184 |
| 2019 | 5 | -0.44 | 0.17 | -0.002 | 0.001 | 238 |
| 2019 | 6 | -0.29 | -0.04 | -0.001 | 0.002 | 1215 |
| 2019 | 7 | 0.33 | 0.22 | -0.001 | 0.001 | 1178 |
| 2019 | 8 | 0.11 | 0.13 | -0.001 | 0.001 | 1123 |
| 2019 | 9 | -0.51 | 0.26 | -0.001 | 0.001 | 680 |

The ratio of the provided sc-AOD and AOD in the ILP output allows us to calculate the corresponding SSA. In the case of ILP retrieved SSA from both Skyrad 4.2 and MRI we can see mainly larger median in Davos (0.952/0.926 for 500/870 nm from



Skyrad 4.2 and 0.959/0.939 from MRI) compared to Rome (0.934/0.917 from Skyrad 4.2 and 0.942/0.927 from MRI). The monthly values are in the supplement table S12. The difference between the 80th and 20th percentiles of the SSA overall is larger in Rome at 500 nm (0.03/0.02 from Skyrad 4.2 at 500/870 nm and 0.025/0.015 from MRI) and larger in Davos at 870

nm (0.021/0.029 from Skyrad 4.2 nm and 0.014/0.02 from MRI). However, there are month to month variations. In the supplement table S13 we show the monthly medians of the daily differences between the 80th and 20th percentiles. Depending on the month either Rome or Davos shows larger variability. The number of available QUATRAM II common measurements is 1114 for Davos and 4434 for Rome.

## 4 Discussion

In section 3.1.1 we compared the AOD between several PFRs and POMs in two locations with different characteristics (Davos and Rome) under different calibration methods of the POM. Using the original POM AOD (calculated after ILP calibration of the POMs) we found that the POMs provide systematically lower AOD than the PFRs. This systematic difference is larger in Rome. Using calibration transfers with the PFR as reference to re-calibrate the POMs we achieved excellent agreement showing that the differences between the post processing algorithms of the networks and the technical characteristics have

only minor effect on AOD differences. The major cause of AOD difference was the calibration method. The calibration differences per campaign were approximately 0.7-1.6% in Davos and 1.6-3.5% in Rome at 500 nm and 0.2-1.8% in Davos and 1-3.4 % in Rome at 870 nm (supplement table S1). The AOD differences per campaign were approximately 0.005-0.01 in Davos and 0.015-0.035 in Rome at 500 nm and 0-0.005 in Davos and 0.005-0.017 in Rome at 870 nm (section 3.1.1).

We also compared the AOD between the reference PFR and the co-located CIMEL for each case for cross-validation. All

median AOD differences between CIMEL and PFR were below 0.01 and the traceability criteria are satisfied with the exception of the Rome phase in QUATRAM I campaign and the 500 nm of Rome phase in QUATRAM II campaign. The generally good agreement between PFR and CIMEL is consistent with the small differences of the CIMEL and PFR based calibration transfers in Campanelli et al., 2023.

Regarding the PFR calibrations the uncertainty is lower as shown in section 3.1.2. The PFRN01 and PFRN14 used for the

Rome phases showed good calibration stability before and after their shipments (section 3.1.2). The PFRN27 used in the Davos phases as a reference was for the whole 2017-2021 period present in Davos as part of the PFR reference triad. Also, it is used in a long-term comparison study with AERONET (Karanikolas et al., 2022) showing very good agreement with CIMEL in the period 2007-2019.

Attempting to explain the observed calibration differences we investigated whether the two stations show some systematic

difference during the campaigns in terms of aerosol properties' values or variability that could explain the different calibration performance. The available datasets of AOD, SSA and AE showed no such association. However, the AERONET SSA dataset has important limitations of data availability and accuracy as explained in section 2.4.1. One explanation could be that the values or the variability of SSA and AE affect the calibration proportionally to the AOD levels. However, we cannot identify





such association as well from our results (details in Fig. 2-4 and supplement table S1). In Davos the last two months of
QUATRAM II (9-10/2018) show similar calibration difference between ILP and calibration transfers under different
conditions in all 3 parameters (AOD, SSA and AE). Also, in QUATRAM I (8/2017) the AOD at 500 nm is above 0.1, while
in QUATRAM III (10/2021) below 0.05, but the calibration difference is smaller in QUATRAM I. In Rome at QUATRAM II
the first month (5/2019) shows simultaneously the lowest AOD and SSA variability in both wavelengths. At 500 nm the second
and fourth months (6 and 8/2019) show smaller calibration difference, while AOD is higher and all three parameters more
variable. The third month (7/2019) shows the largest calibration difference under similar AOD and SSA conditions with 6 and
8/2019, but lower AE variability.

We also conducted a sensitivity analysis of the ILP method under different conditions on its six input parameters (Real part of
refractive index (RRI), Imaginary part of refractive index (IRI), Surface albedo (SA), Total Ozone Colum (TOC), Surface
Pressure (P) and Solid View Angle (SVA) ). SVA and SA errors can explain part of the ILP calibration underestimation.
Regarding IRI the ILP calibration showed very little sensitivity of most months (which is consistent with the study in
Campanelli et al., 2004), but very large for specific months and IRI values showing some evidence of model instabilities under
certain conditions combinations of NSR and IRI values. RRI, TOC and P showed no evidence of significant effect. However,
most part of the calibration differences remained unexplained.

By comparing the retrieved AOD from the Skyrad code with PFR AOD we can identify an underestimation mainly in Rome,
although smaller than the AOD retrieved from direct sun scans and ILP calibration. However, ILP uses sc-AOD instead of
AOD for the calibration. A stronger underestimation of sc-AOD compared to AOD or dependence of the sc-AOD error with
the air mass can explain the calibration difference. Such underestimation may be not fully visible in the AOD dataset due to a
systematic error in the ILP inverted SSA that reduces the AOD error. Using an alternative inversion model (Skyrad MRI) we
found no significant systematic difference of sc-AOD. The ILP calibration using MRI had positive and negative differences
from the original ILP mainly by less than 1%. Such differences can be attributed to the different selection of data and random
differences of sc-AOD between the 2 models. Under both models we found no consistency between the SSA variability
corresponding to the provided sc-AOD/AOD. The AERONET median SSA is higher in Davos (0.02), however, the difference
is within the uncertainty of the inversions and corresponds to different scattering angles. Also, the high SSA uncertainties and
the mainly low sensitivity of the ILP to the imaginary part of the refractive index limit further the significance of this finding.
Another issue related to the ILP calibration is its random uncertainty. Despite the clear systematic bias we observed compared
to the calibration transfers, the random fluctuations remain significant. In section 3.1.2 we showed that there can be both
fluctuations for consecutive months and estimated uncertainties of ILP calibration above 1%. The lack of coincidence between
the month-to-month variability of ILP and transfer-based calibrations suggests that indeed we cannot attribute these
fluctuations to instabilities of the instruments. The calibration transfers showed smaller uncertainty and larger stability apart
from large shifts during specific months. The PFR calibrations are more stable and have smaller uncertainties than the
calibration transfers, so we cannot attribute the calibration transfer fluctuations to changes in the PFR. However, as described
in section 3.1.2, we cannot attribute all ILP fluctuations to the CV% of the ILP calibrations and changes in the instruments,



but rather to the overall ILP uncertainty. A potential source of uncertainty (or bias) is the linearity of the fit during ILP. The currently used linear fitting standard error threshold may allow discrepancy from the linear behaviour large enough to cause

uncertainties at the observed level. More research is needed to further clarify the matter.

The calibration underestimation observed by ILP compared to the calibration transfers is probably a result of errors in the sc-AOD retrievals. As ILP method shows sensitivity mainly to the provided normalized sky radiance (NSR), the retrieval errors are probably a result of assumptions in the forward model that simulates the NSR. The effect is amplified in Rome compared to Davos. A known constant difference between the two locations is the altitude. As Davos is higher (by ~1500 m), the

atmospheric pressure is constantly lower leading to a reduced Rayleigh scattering optical depth, which contributes towards a reduced DSI and decreased multiple radiation scattering. Therefore, the NSR dependence with the scattering angle can be systematically different between the two locations for any given SZA. In that case, the forward model of ILP may simulate less accurately the effect of the multiple scattering in Rome or the increased multiple scattering there may amplify the errors of the simulations. More research is required to investigate whether the source of the larger calibration differences in Rome.is

indeed due to the lower altitude of the Rome station and to what extent it can be generalized for other sites.

Significant improvement seems to be possible using Cross Improved Langley Plot (XILP) (Nakajima et al., 2020; Campanelli et al., 2023) instead, which seems to lead in smaller biases. XILP performs ILP with the axes reversed, but also includes different criteria for the selection of data used for the final linear fit and the days considered as valid. However, XILP also showed a few cases of large differences (or even larger than ILP) compared to the calibration transfer. Therefore, more research

is required to assess the XILP sensitivity in the sc-AOD, inputs parameters and whether it can lead to long-term traceability of AOD regardless the location and the conditions.

**5 Conclusions**

In this study we assess AOD differences between GAW-PFR and ESR instruments and investigate their causes. We used data of three intercomparison campaigns with two phases each. One phase was in Davos, a mountainous area and one in Rome a

low altitude urban area. Comparison of different pairs of PFR and POM instruments showed that the traceability criteria are satisfied at 870 nm in Davos for all campaigns and Rome in one campaign. At 500 nm they are not satisfied, but in Davos the differences are smaller and below the AOD standard uncertainty (median AOD difference below 0.01). Our analysis shows that the contribution of the instrument and post processing differences to the AOD differences is minor. The major cause is the different calibration methods. We concluded that the ILP calibration method used by ESR results to a systematic

underestimation of the calibration constant and as a result the AOD, compared to GAW-PFR and AERONET measurements. Our investigation on the causes showed that part of the difference (mainly at 500 nm) can be explained by potential errors in the surface albedo and the instrument solid view angle used as input for the ILP calibration. However, the largest part of the difference cannot be attributed to errors in the input parameters. It can be explained by errors in the sc-AOD retrieval, which is required to perform ILP. The error is probably a result of the forward model assumptions. A potential explanation could be





related to the way the model handles multiple scattering, which probably amplifies the error in lower altitude sites. This work is a demonstration of the limitations and challenges of the ILP 'on-site' calibration procedure for sun photometers. The present study and Campanelli et al., 2023 offer a starting point for future research to their further understanding towards more general conclusions and potential improvements.

*Code availability*. The used version of SKYRAD 4.2 code package is available through communication with the authors.

*Data availability*. The CIMEL AOD data are available from https://aeronet.gsfc.nasa.gov/

The PFR and POM raw signals and AOD data are available through communication with the authors.

*Author contribution*. AK analysed the data and wrote the paper with contributions from the co-authors. AK and SK conceptualized the study. NK and SK contributed to the PFR sun photometer data provision. NK assisted with the CIMEL and PFR sun photometer data selection. MC and VE contributed to the POM sun and sky radiometer data provision. MC, MM and GK contributed to the SKYRAD 4.2 pack code provision and assisted with its operation. GK contributed with the SKYRAD pack MRI output. All authors were involved in the interpretation of the results and reviewing the paper.

*Competing interests*. The authors declare that they have no conflict of interest.

*Acknowledgments*. This work has been supported by the European Metrology Program for Innovation and Research (EMPIR) within the joint research project EMPIR 19ENV04 MAPP "Metrology for aerosol optical properties". The EMPIR is jointly funded by the EMPIR participating countries within EURAMET and the European Union.

Stelios Kazadzis would like to acknowledge the ACTRIS Switzerland project funded by the Swiss State Secretariat for Education, Research and Innovation.

The participation of Gaurav Kumar has been also supported by the Spanish Ministry of Economy and Competitiveness and the European Regional Development Fund through project PID2022-138730OB-I00, and Santiago Grisolia program fellowship GRISOLIAP/2021/048.



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
