# Peer review of "Intercomparison of AOD retrievals from GAW-PFR and SKYNET sun photometer networks and the effect of calibration"

_Atmospheric Measurement Techniques, 2024_

## Referee Comment (RC1)

**Referee comment on "Intercomparison of AOD retrievals from GAW-PFR and SKYNET sun photometer networks and the effect of calibration" by Karanikolas et al.**

Anonymous Referee

This paper describes the intercomparison analysis between the AOD retrieved from the three different photometer models currently used in the most important aerosol monitoring networks. This is a quite interesting topic which has not been addressed in this level of detail so far in the literature. Therefore, the results presented here can be considered the reference publication to understand the reasons behind the different performance of these instruments. This topic is not only challenging but crucial when it comes to using these different datasets for global climate studies.

This referee believes that the manuscript aligns perfectly with the scope of AMT, and the presented results are indeed relevant for publication in this journal. However, there are some general comments that this referee would like to see addressed before final publication, in addition to some technical-minor remarks.

**General comments:**

The use of English is poor. I strongly encourage the authors to have the paper checked by a native English speaker. There are grammar problems which make the paper quite difficult to read and understand in some parts.

In general, the paper includes many aspects impacting the accuracy of the POM AOD product. The inclusion of all these factors (very challenging task) is a strong point of this study, resulting in a comprehensive review of the factors affecting this product. However, the way in which this large amount of information is presented is truly complicated, and if not expressed correctly, it can result in a study that is not properly understood and does not adequately link the different sections. For this reason, I encourage the authors to try to better organize or somewhat synthesize the results presented here.

**Minor/Technical comments:**

References: Please check the reference format. Many references in the text do not follow the journal's requirements.

Decimal places: Please standardize the decimal places used in the current work according to the journal's requirements.

Numbering: Please ensure that all numbering used in the current work follows the journal's requirements.

Capital letters: Please standardize the capitalization of terms such as Sun photometer, sun photometer, Sun-sky photometer, etc.

Abstract, Line 15: Correct use of commas: "… Improved Langley calibration method, (ILP), used by SKYNET, and…".

Line 17: Please correct this sentence: One is a mountainous rural area (Davos, Switzerland) and the other is urban (Rome, Italy).

Abstract, Line 18: Is "where" the proper wording in this case?

Abstract, Line 21: Is "In Rome at 500 nm…" a proper wording?

Abstract, Line 33: In "input parameters needed for it" please define what "it" is. Additionally, please correct: "… we report on **the** results **of** the AOD retrievals".

Line 44: I'm wondering if the authors can include the latest version (AR6) of the IPCC (2023).

Line 67: Do the authors consider Doppler et al. (2023) to be the only reference for intercomparison campaigns between photometers?

Lines 94 and 95: The authors refer to Table 1 in this part of the paper, which is located in a different section.

Line 97: Why emphasize at this point that it is a discussion paper? By the time the paper is published, it might no longer be in the discussion stage, making this paper outdated...

Line 100-101: Please rephrase; the extra comma and the final part make the sentence unclear to the reader.

Line 104: "The data used" is referred to this work/paper? Sometimes I feel there is a lack of information to understand the sentences. The entire sentence is not clear to me.

Line 109-110: Something is missing in this sentence.

Line 110: **a** co-located Cimel?

Line 111: I don't understand what "initial" means in this sentence. Please rephrase.

Table 1: The mention to the ability to perform lunar measurements is only mentioned in this table. Why not mention this in Sect. 2.1.2?

Line 145: FWHM is different for 1640 nm spectral band. Please correct.

Line 149: Please delete "The" before AERONET AOD data.

Line 155: Please correct the typo in "Langley".

Lines 171-172: Is there a typo at the end of the sentence? I'm not able to understand it.

Line 181: Please correct "**to** increase" and "**their** status".

Line 182: I consider it is more useful to include that it is usable at every type of station.

Line 184: Please delete the extra point.

Lines 185-185: Some commas could improve the readability of this sentence.

Line 199: **To** retrieve…

Line 201: It is strange using "also" in a new paragraph…

Line 204: What is the "case" you are referring?

Lines 204-206: Please rephrase.

Line 211: Add "the" before POM.

Lines 211-212: Please rephrase: "The calibration constants and raw signals are in the units measured by each instrument and are corrected ..."

Line 214: **A** diurnal variation seems to be the reason for restricting ratios. Is it something that happens every day? If so, "**A**" is not necessary.

Line 215: Time interval?

Line 216: Rest of data? This sentence seems unfinished...

Line 216: 2 std of the points? Please correct.

Lines 216-217: I don't understand this sentence…

Lines 217-218: Is the same criterion than the stated in line 215?

Line 218: Is this rejection regarding calibration error a visual analysis or it is based on a certain threshold?

Lines 223-224: I'm not sure I've understood correctly this sentence. Between 2 months the authors use a linear interpolation between two values, with a constant value during the whole month. But what happens in the case of the first month of measurements?

Line 225: I suggest to remove this sentence: "The actual wavelength of each instrument may vary".

Line 238: The authors are talking about two different datasets: original ESR and calculated from calibration transference. In this sentence, the authors are stating that the second dataset (own calibration) does not include NO2 correction but SKYNET (first dataset) includes it? Please explain why.

Line 243: Please add reference for the AERONET algorithm.

Line 244: Are the authors using a double screening? AERONET & PFR cloud screenings.

Line 252: I understand what the authors are pointing here, but don't you think that anticipating the reader with results which will be published later in the paper is not recommendable? Maybe the authors can reference a published paper (Campanelli, 2023) or move this section to the results section?

Line 257: Is the first time that tau_sc is written as sc-AOD? I think it is better to standardize the terms to prevent the reader from getting lost when reading the paper.

Line 261: This is a general comment. Don't you think that the sentence "There are three parameters included in this section" has perfect meaning without the words "which be"? In my view, the use of these extra words leads to a wordy reading.

Line 264: Its variability? Variability of what variable?

Line 267: Is there an estimation on the different uncertainty of levels 1.5 and 2.0 or this sentece is relating large uncertainty and the lack of QA/QC of AERONET level 1.5 data?

Line 268: change **in**? (Repeated twice in this sentence).

From line 268 onwards: From this point onward, this referee will no longer correct grammar-related aspects as it is not the object of the review and I am not a native English speaker. I would like to recommend that the authors undertake a thorough correction of the writing and the English used in this manuscript. It is difficult to understand some reasoning presented in this work.

Line 277: Are these abbreviations already included before in the text? If so, please make use of abbreviations.

Line 288: Since the restriction to QUATRAM-II campaign is presented in this specific paragraph, focus on SVA, this referee understand that these input files are only referred to SVA and not to the rest of parameters (P, SA, etc). Can the authors confirm?

Line 289: Since the authors will follow the same structure in section 3.2.2 I would recommend separate these sub-studies more clearly in this section, according to the sub-sub-sub-sections defined in section 3.2.2.

Line 299: Do the authors mean: in the case of SA? I don't understand what the authors are referring here.

Line 324: The method uses log(DSI) **plus Rayleigh and gas absorption terms** versus m*AOD_sc, right?

Line 346: The methods presented in this paper related to POM retrievals seem to be more accurate at high AOD conditions but then, cloud screening impose AOD to be below 0.4. I understand that this assumption is out of the scope of this paper, but I'm wondering if the authors could explain shortly in plain words if you see any inconsistency between these criteria or if you expect to change them in the future.

Line 369: Yes, it is quite noticeable the differences observed during this campaign, especially at 500 nm. Are these results already published or the authors have any explanation for these high discrepancies?

Line 384: I don't understand "random differences within the retrieved uncertainty". Can the authors elaborate what this partial conclusion means?

Figure 1: What are the two red lines at ±0.01? It will be useful here include "(POM_IL-PFR)", as it is stated in the legend. The same for "POM_TR" and "Cimel-PFR".

Table 2: I suggest to define the instruments as POM-XXX or POM/XXX to make clear the two words (instrument + location). I know that you have also the difference in calibration method (POM_XX) that can cause confusion...

Lines 407-409: Please rephrase this sentence. The part in parenthesis is a sentence itself and it is very difficult to read as it is.

Line 413: Is it expected the uncertainties of ILP to be purely random?

Line 413: Is the estimation "evident"?

Line 424: What are the fluctuations expressed here? Are authors referring to the amplitude of the uncertainties previously reported? I'm not able to understand this paragraph...

Line 449: Can you please help the reader with a reference to the corresponding figure, table or number?

Figure 2: AOD in the figure caption is referred to PFR, right? Can you clarify?

Line 466: Can you please help the reader with a reference to the corresponding figure, table or number? The same for the rest of partial conclusions.

Line 471: Is 2.4.1. a reference to a Section?

Figure 3: Is the SSA extracted from AERONET? Please clarify.

Line 481 and Figure 4: Is the AE retrieved from PFR? Please clarify.

Line 496: Is this analysis restricted to QUATRAM_II campaign as stated in line 289?

Line 502: As commented in line 287, the name of these sub-studies should be similar and clearly related to the description in line 287.

Line 514: Are the acronyms already defined? If so, use the acronym.

Line 518: Is not all this section referred to QUATRAM-II?

Line 530: What type of modifications the authors expect in the v4.2?

Figure 5: Do the authors think that the title of the graphs are needed? The same for the rest of figures. I suggest to include panels (a) and (b) rather than "left side" and "right side" in the caption. The same for the rest of figures.

Table 4: The first sentence in the caption case seem unfinished. What "selected case" means?

Figure 8: Again, comment on title and legend meaning.

Line 615: I recall reading in the first part of the paper about the difference between Skyrad and MRI and the importance of including the second method. However, unfortunately, after so much information, at this point, the reader no longer remembers that information. This comment is not aimed at repeating the information in this section but simply to inform the authors that reading this paper, as it is presented, is quite challenging.

Table 6: As in the case of the figures and previous tables, in the caption should appear the information as written in the table. In this case, I believe that (P5th, P95th) should be mentioned.

Table 7: The same for Delta V0.

Line 662: Maybe it is interesting to provide here some numbers about the systematically lower AOD?

Line 681 onwards: Please, focus only on the most important information...

Line 693: Is it necessary to repeat again the acronyms?

Line 745: Can the authors provided here some numbers when they talk about the underestimation?

---

## Author Comment (AC1)

Author's response to referee #3

You can find our response below each comment.

*General comment about language.*

We included another co-author, who assisted with a large number of corrections in terms of English grammar, syntax and expression.

*The authors use many acronyms in the document. A table summarizing the acronyms and their definitions would be useful for the readers.*

Added a table of acronyms in the supplement.

*I also recommend that the authors should write a few lines (e.g., in the introduction) about the novelty and the usefulness of their study.*

Added at the end of the introduction:

"In previous studies intercomparisons (eg Kazadzis et al., 2023), the study of AOD differences was limited to the differences of AOD provided by each network. In the present study, we also separate the effect of the calibration approaches and the effect of the post-processing and instrument differences. We also include one campaign at each location with a duration of several months, which provided a significantly larger amount of data compared to the shorter campaigns that are more frequently organized. Finally, we include a detailed analysis of the ILP calibration method in relation to the aerosol properties and its sensitivity to all required input parameters."

*L18: "The major". Do the authors mean "The main"? Furthermore, while the authors write here that AOD uncertainties are mostly due to the calibration method, in lines 22-23 they write that they did not find any association between the calibration performance and the variability in aerosol properties. This is confusing.*

Changed to "main".

The following are 2 separate issues. One finding is that the AOD uncertainties are mostly due to the calibration differences. Another finding is that we did not find any association between the calibration performance and the variability in aerosol properties. Probably the second finding is not understandable without the context in the main text, so we removed this sentence.

*L19: "Underestimation of AOD compared to GAW-PFR". The AOD from which network?*

This refers to the ESR AOD. Rephrased to "underestimation of ESR AOD compared to GAW-PFR".

*L20: "Underestimation in the ILP calibration". What is underestimated?*

This refers to the systematically lower calibration constant (extraterrestrial signal) retrieved by ILP compared to the transferred calibrations using the PFR as reference. Rephrased to:

"underestimation of the calibration constant calculated with the ILP method compared to the calibration transfers using PFR as reference".

*L23: "variability in" instead of "variability of"?*

No longer applicable.

*L28-29: "between … calibrations". Do the authors mean the AOD retrieved the two methods?*

The sc-AOD retrieved by the two methods and the calibration constant each case leads to. Rephrased to:

"between the retrieved sc-AOD nor a systematic increase in the ILP derived calibration constant when using the MRI pack for sc-AOD inversion instead of the Skyrad 4.2".

*L31: "AOD retrieved using ILP" instead of "ILP"*

This sentence refers to the calibration methods (ILP and standard Langley) not the AOD.

*L40: The phrase "the major driver" is not correct. It should be either "the main driver" or "a major driver".*

Replaced with "the main driver".

*L43: Delete "a".*

Reformulated to:

"can lead to  a significant forcing"

*L47-48: "AOD is … of aerosols". Please rephrase.*

Rephrased from:

 "AOD is an indicator of the total aerosol load in the atmosphere and its spectral

dependence with the size of aerosols."

to:

 "AOD describes the overall effect of the total aerosol column on the attenuation of solar radiation, and is correlated with the total aerosol load in the atmosphere and its spectral dependence with the size of aerosols."

*L63: "Due to the differences …". Differences in what?*

The differences are described in the previous 6 sentences. Rephrased to:

"Due to the differences among the main networks (i.e., AERONET, GAW-PFR, SKYNET) described above".

*L66: "of all types" instead "from all types".*

Rephrased to:

"includes all types of sun photometers".

*L69: "that the instrument" instead of "the instrument".*

Corrected.

*L70-73: Please re-write clearer. Furthermore, the next sentence ("The conventionally … instruments") repeats the same things that the authors write in lines 70-73.*

Rephrased from:

"There are different ways to calibrate a sun photometer. It can be accomplished either by using a co-located instrument as a reference, by laboratory calibration to the international system of units (SI) and use of satellite measurements for the top-of-the atmosphere or by using an indirect method to calibrate the instrument through the DSI at the ground. The conventionally used methods are the standard Langley plot method (SLP) (Shaw et al., 1973) and the calibration transfer from a reference instrument."

to:

"There are different ways to calibrate a sun photometer. Conventionally, they are calibrated by the standard Langley plot method (SLP) (Shaw et al., 1973) and the calibration transfer from a reference co-located instrument. An alternative method is the laboratory calibration to the international system of units (SI). Under this alternative approach, we can use satellite measurements for the top-of-the atmosphere irradiance that are also in SI units."

*L85: Replace "One of the main differences are" with "One of the main differences between … is".*

Rephrased to:

"One of the main differences between GAW-PFR with AERONET and SKYNET is correction for absorption due to nitrogen dioxide (NO2) and water vapor (H2O)".

*L91: "a Memorantum of Understanding" instead of "and Memorantum of Understanding"*

Corrected.

*L96: "using both," instead of "both with".*

No longer applicable.

*L118: "centered at" instead of "centered on"?*

Corrected.

*L141: "or diffuse sky radiance" instead of "and diffuse sky radiance"*

Corrected.

*L226: "the difference … 1-2 nm". Is there any reference that can be used to support this statement? Can the authors explain why the difference can be up to 2 nm?.*

This was based on the observed differences of the central wavelength between the nominal wavelength and the one taken from the characterisation of the PFRs and CIMELs (available in the AOD files of those instruments). We removed this sentence from the manuscript.

*L235: "differ in" instead of "differ on".*

Corrected.

*L253: Larger than what?*

Larger than in Davos. Rephrased to "larger in Rome compared to Davos".

*L266: add "were used" after "retrievals".*

Rephrased "For the SSA the AERONET level 1.5 retrievals"

 to

"For the SSA, we used the AERONET level 1.5 retrievals".

*L282: Is there any reference for Eq. 8?*

This equation was not taken from a particular literature reference. It's the standard equation used by ESR.

*L292-293: The manuscript is not clear at this point. Do the authors mean that the TOC from OMI overpasses is used in the PFR algorithm?*

Yes. Rephrased from:

 "TOC is taken OMI overpass (aura_omi_l2ovp_omto3_v8.5 https://acd-ext.gsfc.nasa.gov/anonftp/toms/omi/data/overpass/) "

to:

 "The TOC used in the PFR algorithm corresponds to the OMI satellite product (aura_omi_l2ovp_omto3_v8.5 https://acd-ext.gsfc.nasa.gov/anonftp/toms/omi/data/overpass/)".

*L332: "number of retrievals" instead of "retrievals"*

Corrected.

*L366: "II and 3": please be consistent with the numbering*

Corrected.

*L373: Delete in that case"*

Deleted.

*L375: Using the same numbering in the supplement and in the main text is confusing. Please change the format of the numbering in the figures and tables in the supplement (e.g., Fig. S1, Fig S2, …, Table S1, …).*

Corrected.

*L407: "due to" instead of "due".*

Corrected.

Yes. Rephrased to:

"instruments' response".

Rephrased from:

"An estimation of the uncertainty magnitude is evident in the coefficient of variation (CV%) of the daily ILP calibrations per month (Campanelli et al., 2023 preprint table 2a) which are between 0.18%-2.87% at 500 and 870 nm."

to:

"The coefficient of variation (CV%) of the daily ILP calibrations per month (Campanelli et al., 2023 table 2a) is an estimate of the ILP monthly calibration uncertainties. The CV% for the ILP calibrations used in this study range between 0.18%-2.87% at 500 and 870 nm."

Figures 2-4: It would help to somehow show the months at which each box corresponds

Added the months in a second horizontal axis (upper part of the graph).

Rephrased from:

"ILP assumes a constant SSA as the inverse slope the linear fit (section 2.2.1) and the refractive index pre-assigned to specific value which potentially reduces the accuracy of the method. Here we present the AERONET SSA values and variability between the months of the campaigns (Fig. 3) at 440 nm (green) and 870 nm (red)."

to:

"The ILP method assumes a constant SSA as the inverse slope of the linear fit (section 2.2.1) and uses an a-priori refractive index (selected by the operator). These assumptions potentially reduce the accuracy of the method. Here we present the SSA values provided by AERONET and their variability during the campaign months (Fig. 3) at 440 nm (green) and 870 nm (red)."

Rephrased from:

"Similarly, in Davos the largest variability in during QUATRAM III (DAV21), which also exceeds the Rome SSA variability."

to:

"Similarly, the largest variability is during QUATRAM III (DAV21) in Davos, which also exceeds the Rome SSA variability."

*L483: As an additional indicator?*

Added "additional".

*L511: "for both locations" instead of "both locations"*

Corrected.

*L513-514: Delete "and"*

Replaced (Sections and tables S8-S10) with (Tables S8-S10)

*L561: "% difference" instead of "%difference"*

Corrected.

*L562: and the transferred calibration?*

Yes, added "transferred".

*L572: Delete "be".*

Corrected.

*L578: "At … Rome": Please rephrase.*

Rephrased from:

"At 870 nm the difference remains at least for 0.95%

Davos and 1.7% for Rome."

to:

"At 870 nm the difference is at least 0.95% for Davos and 1.7% for Rome for all SA values used as input."

*L582: "at 870 nm it is" instead of "at 870 nm".*

Rephrased to "at 870 nm, results are".

*L606: "that there is a" instead of "the significance of the"*

The systematic bias of the ILP calibration was clear already earlier (AOD and monthly calibrations). Here we show to what extent this appears to the daily ILPs (1 or 2 per day). The significance refers to the frequency of the bias' appearance in the daily data.

Deleted this short sentence from the manuscript.

*L618:"and thus it may show" instead of "it may show".*

Replaced "it" with "and".

*L625: "-0.004" instead of "-004"?*

Corrected.

*L716: in the PFR performance?*

Rephrased to "changes in the PFR response".

Corrected.

Significant modifications not requested by this referee:

As a response to the general comments of another referee we changed the order of the sections to improve the readability of the manuscript.

The methodology and the results are no longer the separate sections 2 and 3. Section 2 will now include only the parts of the preprint section 2 up to section 2.2.2. Each part of methodology starting from the preprint section 2.3 (methodology of the different sub-studies), will be now followed by the corresponding results directly in the next sub-section.

The modifications are the Table R1 below. We show each section of the pre-print to which section will correspond of the revised manuscript.

Table R1: The difference between the structure of the manuscript between the preprint and the revised version.

| Preprint sections | Revised sections | Comments | Subject |
|---|---|---|---|
| 2.3 | 3 | 3.1 methodology, 3.2 Results | Intercomparisons |
| 3.1.1 | 3.2.1 | - | Intercomparisons of AOD for different calibrations |
| 3.1.2 | 3.2.2 | - | Uncertainties |
| 2.4 | 4 | - | ILP error sources |
| 2.4.1 | 4.1 | 4.1.1 methodology 4.1.2 results | Aerosol properties |
| 3.2.1 | 4.1.2 | - | Aerosol properties-results |
| 3.2.2 | 4.1.2.1 | - | AOD |
| 3.2.3 | 4.1.2.2 | - | SSA |
| 3.2.4 | 4.1.2.3 | - | AE |
| 2.4.2 | 4.2 | 4.2.1, 4.2.2 and 4.2.3 the 3 sensitivity sub-studies. 4.2.1.1, 4.2.2.1 and 4.2.3.1 the methodology of each, previously merged in section 2.4.2. Added "Sub-study 1,2 or 3: in the corresponding titles (including the result sections below). | Sensitivity of ILP to input parameters |
| 3.2.2.1 | 4.2.1.2 | - | Results: Test one input as variable per case |
| 3.2.2.2 | 4.2.2.2 | - | Results: All input parameters variables |
| 3.2.2.3 | 4.2.3.2 | - | Results: Sensitivity tests |

| 2.4.3 | 4.3 | 4.3.1 methodology
4.3.2 results | AOD and sc-AOD from sky radiance measurements |
|---|---|---|---|
| 3.2.3 | 4.3.2 | - | Results: AOD and sc-AOD from sky radiance measurements |
| 4 | 5 | - | Discussion |
| 5 | 6 | - | Conclusions |

We also removed some sentences from the results and discussion that are not included in the minor comments of any referee to reduce the amount of information presented and fit the new format.

List of the deleted sentences:

1) l.: 34 "In the following sections we report on results on the AOD retrievals of several instruments in different environments using different principles in their calibration methods. We also perform an investigation to explain the causes of differences."

2) l. 94: "During the period 2017-2021 a PFR was transported to Sapienza University in Rome, Italy once for each campaign for several weeks or months to measure AOD in parallel with one or more POMs and CIMEL (Table 1). Also, at least one POM was transported to Davos on 3 different periods as well (Table 1), where the WMO AOD reference (PFR-Triad) and a CIMEL are operated. The POMs were calibrated both with the ILP method and by calibration transfer using a PFR as a reference. There is already a publication under review showing calibration differences between several calibration methods (Campanelli et al., 2023)." (added the period 2017-2021 to the previous sentence)

3) l.:362 "Most of the times in the case of calibration transfer the median difference remains negative, but there are exceptions"

4) l.:434 "As shown in section 3.1.1 the ESR dataset shows a systematic AOD underestimation compared to GAW-PFR and AERONET due to an underestimation in the calibration from the ILP method. However, this calibration difference varies significantly between the two locations and from month to month. Using the methods described in section 2.4 we attempted to explain why this underestimation happens and why it is systematically larger for Rome."

5) l.: 439 "Here we investigate whether there is any systematic difference between Davos and Rome on AOD, SSA and AE values or variability that could potentially be associated with the larger calibration differences in Rome for all months. We use AOD and AE from the PFR data during the half/full days of the ILP calibrations and SSA is from the AERONET data during the QUATRAM campaigns. We used monthly medians as the average level and monthly medians of the daily percentiles (5th,20th, 80th and 95th) as variability indicator as described in section 2.4.1."

6) l.:496 "As the available aerosol conditions during the campaigns show no indication of an explanation to the ILP underestimation and the differences between locations, we attempted to investigate the causes through a sensitivity study of the ILP. ILP uses six parameters as inputs: Real part of refractive index (RRI), Imaginary part of refractive index (IRI), Surface

albedo (SA), Total Ozone Colum (TOC), Surface Pressure (P) and Solid View Angle (SVA). The first five are pre-selected and the last is provided by an in-situ calibration method. Therefore, there are discrepancies between the real atmospheric conditions under which the ILP is performed and the selected values."

7) l.:513 "Due to the small sensitivity at these three parameters, we do not include a more detailed analysis on them, but the comparisons are available in the supplement (sections and tables S8-S10). For the imaginary part of refractive index (IRI), surface albedo (SA) and solid view angle we observed cases of larger sensitivity.

In the Fig. 5-7 we can see the calibration differences between ILP runs and the calibration transfer from PFR for different conditions. The results correspond to the first sub-study described in section 2.4.2 where we study each parameter separately according to the observations of each site. The results correspond to all months of QUATRAM II."

8) l.:631 "Starting from the third column we show the median of all AOD differences, the percentage of differences within the WMO limits, the 5th and the 95th percentiles of AOD differences and the total number of measurements compared per location."

9) l.:638 "The sc-AOD median differences are negative at 500 nm and positive at 870 nm, which is in accordance with the sign of the calibration differences for most cases."

10) l.:686 "Also, in QUATRAM I (8/2017) the AOD at 500 nm is above 0.1, while in QUATRAM III (10/2021) below 0.05, but the calibration difference is smaller in QUATRAM I."

11) l.:717 "and changes in the instruments, but rather to the overall ILP uncertainty"

---

## Author Comment (AC2)

Author's response to referee #2

You can find our response below each comment. We start we a response to the general comments and then we respond to each of the minor comments.

*General comments*:

We included another co-author, who assisted with a large number of corrections in terms of English grammar, syntax and expression.

We also changed the order of sections to improve the readability and flow. The methodology and the results are no longer the separate sections 2 and 3. Section 2 will now include only the parts of the preprint section 2 up to section 2.2.2. Each part of methodology starting from the preprint section 2.3 (methodology of the different sub-studies), will be now followed by the corresponding results directly in the next sub-section.

The modifications are the Table R1 below. We show each section of the pre-print to which section will correspond of the revised manuscript.

Table R1: The difference in the structure of the manuscript between the preprint and the revised version.

| Preprint sections | Revised sections | Comments | Subject |
|---|---|---|---|
| 2.3 | 3 | 3.1 methodology, 3.2 Results | Intercomparisons |
| 3.1.1 | 3.2.1 | - | Intercomparisons of AOD for different calibrations |
| 3.1.2 | 3.2.2 | - | Uncertainties |
| 2.4 | 4 | - | ILP error sources |
| 2.4.1 | 4.1 | 4.1.1 methodology 4.1.2 results | Aerosol properties |
| 3.2.1 | 4.1.2 | - | Aerosol properties-results |
| 3.2.2 | 4.1.2.1 | - | AOD |
| 3.2.3 | 4.1.2.2 | - | SSA |
| 3.2.4 | 4.1.2.3 | - | AE |
| 2.4.2 | 4.2 | 4.2.1, 4.2.2 and 4.2.3 the 3 sensitivity sub-studies. 4.2.1.1, 4.2.2.1 and 4.2.3.1 the methodology of each, previously merged in 2.4.2. Added "Sub-study 1,2 or 3: in the corresponding titles (including the result sections below). | Sensitivity of ILP to input parameters |
| 3.2.2.1 | 4.2.1.2 | - | Results: Test one input as |

| | | | variable per case |
|---|---|---|---|
| 3.2.2.2 | 4.2.2.2 | - | Results: All input parameters variables |
| 3.2.2.3 | 4.2.3.2 | - | Results: Sensitivity tests |
| 2.4.3 | 4.3 | 4.3.1 methodology 4.3.2 results | AOD and sc-AOD from sky radiance measurements |
| 3.2.3 | 4.3.2 | - | Results: AOD and sc-AOD from sky radiance measurements |
| 4 | 5 | - | Discussion |
| 5 | 6 | - | Conclusions |

As a response to another referee's comment, we also include a table of acronyms in the supplement.

Finally, we removed some sentences from the results and discussion that are not included in the minor comments to reduce the amount of information presented and fit the new format.

List of the deleted sentences:

1) l.: 34 "In the following sections we report on results on the AOD retrievals of several instruments in different environments using different principles in their calibration methods. We also perform an investigation to explain the causes of differences."

2) l. 94: "During the period 2017-2021 a PFR was transported to Sapienza University in Rome, Italy once for each campaign for several weeks or months to measure AOD in parallel with one or more POMs and CIMEL (Table 1). Also, at least one POM was transported to Davos on 3 different periods as well (Table 1), where the WMO AOD reference (PFR-Triad) and a CIMEL are operated. The POMs were calibrated both with the ILP method and by calibration transfer using a PFR as a reference. There is already a publication under review showing calibration differences between several calibration methods (Campanelli et al., 2023)." (added the period 2017-2021 to the previous sentence)

3) l.:362 "Most of the times in the case of calibration transfer the median difference remains negative, but there are exceptions"

4) l.:434 "As shown in section 3.1.1 the ESR dataset shows a systematic AOD underestimation compared to GAW-PFR and AERONET due to an underestimation in the calibration from the ILP method. However, this calibration difference varies significantly between the two locations and from month to month. Using the methods described in section 2.4 we attempted to explain why this underestimation happens and why it is systematically larger for Rome."

5) l.: 439 "Here we investigate whether there is any systematic difference between Davos and Rome on AOD, SSA and AE values or variability that could potentially be associated with the larger calibration differences in Rome for all months. We use AOD and AE from the PFR data during the half/full days of the ILP calibrations and SSA is from the AERONET data during the QUATRAM campaigns. We used monthly medians as the average level and monthly medians of the daily percentiles (5th,20th, 80th and 95th) as variability indicator as described in section 2.4.1."

6) l.:496 "As the available aerosol conditions during the campaigns show no indication of an explanation to the ILP underestimation and the differences between locations, we attempted to investigate the causes through a sensitivity study of the ILP. ILP uses six parameters as inputs: Real part of refractive index (RRI), Imaginary part of refractive index (IRI), Surface albedo (SA), Total Ozone Colum (TOC), Surface Pressure (P) and Solid View Angle (SVA). The first five are pre-selected and the last is provided by an in-situ calibration method. Therefore, there are discrepancies between the real atmospheric conditions under which the ILP is performed and the selected values."

7) l.:513 "Due to the small sensitivity at these three parameters, we do not include a more detailed analysis on them, but the comparisons are available in the supplement (sections and tables S8-S10). For the imaginary part of refractive index (IRI), surface albedo (SA) and solid view angle we observed cases of larger sensitivity.

In the Fig. 5-7 we can see the calibration differences between ILP runs and the calibration transfer from PFR for different conditions. The results correspond to the first sub-study described in section 2.4.2 where we study each parameter separately according to the observations of each site. The results correspond to all months of QUATRAM II."

8) l.:631 "Starting from the third column we show the median of all AOD differences, the percentage of differences within the WMO limits, the 5th and the 95th percentiles of AOD differences and the total number of measurements compared per location."

9) l.:638 "The sc-AOD median differences are negative at 500 nm and positive at 870 nm, which is in accordance with the sign of the calibration differences for most cases."

10) l.:686 "Also, in QUATRAM I (8/2017) the AOD at 500 nm is above 0.1, while in QUATRAM III (10/2021) below 0.05, but the calibration difference is smaller in QUATRAM I."

11) l.:717 "and changes in the instruments, but rather to the overall ILP uncertainty"

Further sentences are removed or rephrased as a response to the minor comments below.

*References: Please check the reference format. Many references in the text do not follow the journal's requirements.*

Corrected.

*Decimal places: Please standardize the decimal places used in the current work according to the journal's requirements.*

Corrected.

*Numbering: Please ensure that all numbering used in the current work follows the journal's requirements.*

Corrected.

*Capital letters: Please standardize the capitalization of terms such as Sun photometer, sun photometer, Sun-sky photometer, etc.*

Corrected.

*Abstract, Line 15: Correct use of commas: "… Improved Langley calibration method, (ILP), used by SKYNET, and…".*

Corrected.

*Line 17: Please correct this sentence: One is a mountainous rural area (Davos, Switzerland) and the other is urban (Rome, Italy).*

Corrected.

*Abstract, Line 18: Is "where" the proper wording in this case?*

Rephrased to "The main factor leading to AOD differences is the calibration method. We found a systematic underestimation of ESR AOD compared to GAW-PFR due to underestimation of the calibration constant calculated with the ILP method compared to the calibration transfers using the PFR as a reference".

*Abstract, Line 21: Is "In Rome at 500 nm…" a proper wording*

Rephrased to "In Rome the AOD median differences at 500 nm were in the 0.015 - 0.035 range"

*Abstract, Line 33: In "input parameters needed for it" please define what "it" is. Additionally, please correct: "… we report on the results of the AOD retrievals".*

Rephrased to "required input parameters".

The second proposed correction is no longer applicable.

*Line 44: I'm wondering if the authors can include the latest version (AR6) of the IPCC (2023).*

Updated the reference and the citation.

IPCC: Climate Change 2023: Synthesis Report. Contribution of Working Groups I, II and III to the Sixth Assessment Report of the Intergovernmental Panel on Climate Change [Core Writing Team, Lee, H. and Romero, J. (eds.)]. IPCC, Geneva, Switzerland, 184 pp., doi: 10.59327/IPCC/AR6-9789291691647, 2023

*Line 67: Do the authors consider Doppler et al. (2023) to be the only reference for intercomparison campaigns between photometers?*

Doppler et al. 2023 just summarises a large number of related campaigns. We added more references from primary studies about intercomparison campaigns including AOD retrieved from solar irradiance measurements. (Mitchell & Forgan, 2003; Cachotto et al.,2009; Mazzola et al. 2012; Nyeki et al., 2013; Kazadzis et al.,2018a; Gröbner et al., 2023).

Additional references to the pre-existing:

Mitchell, R. M.; Forgan, B. W.: Aerosol Measurement in the Australian Outback: Intercomparison of Sun Photometers. J. Atmos. and Ocean. Tech., 20 (1), 54–66, https://doi.org/10.1175/1520-0426(2003)020<0054:AMITAO>2.0.CO;2, 2003.

Cachorro, V. E., Berjon, A., Toledano, C., Mogo, S. N., Prats, A. M., De Frutos, J. Vilaplana, M., Vilaplana, J. M., Sorribas, M., De La Morena, B. A., Gröbner, J., Laulainenet, N.: Detailed aerosol optical depth intercomparison between Brewer and Li-Cor 1800 spectroradiometers and a Cimel sun photometer, J. Atmos. and Ocean. Tech., 26, no. 8: 1558-1571, https://doi.org/10.1175/2009JTECHA1217.1, 2009.

Nyeki, S., Gröbner, J., Wehrli, C.: Ground-based aerosol optical depth inter-comparison campaigns at European EUSAAR super-sites, AIP Conference Proceedings. Vol. 1531. No. 1. American Institute of Physics, 2013.

Mazzola, M., , Stone, R.S., Herber, A., Tomasi, C., Lupi, A., Vitale V., Lanconelli, C., Toledano, C., Cachorro V.E., O'Neill, N.T., Shiobara, M., Aaltonen, V., Stebel, K., Zielinski, T., Petelski, T., Ortiz de Galisteo, J.P., Torres, B., Berjon, A., Goloub, P., Li, Z., Blarel, L., Abboud, I., Cuevas, E., Stock, M., Schulz, K., H., Virkkul, A.; "Evaluation of sun photometer capabilities for retrievals of aerosol optical depth at high latitudes: The POLAR-AOD intercomparison campaigns." Atmos. Environ., 52, 4-17, https://doi.org/10.1016/j.atmosenv.2011.07.042, 2012.

*Lines 94 and 95: The authors refer to Table 1 in this part of the paper, which is located in a different section.*

No longer applicable.

*Line 97: Why emphasize at this point that it is a discussion paper? By the time the paper is published, it might no longer be in the discussion stage, making this paper outdated...*

The cited paper was indeed accepted for final publication in the meantime, so we corrected the text accordingly.

*Line 100-101: Please rephrase; the extra comma and the final part make the sentence unclear to the reader.*

Rephrased from:

"Also, investigate the extent to which different factors such as the atmospheric conditions and the input parameters required to perform ILP, contribute to the calibration and as a result to retrieved AOD differences."

to

"In addition, we investigate the extent to which different factors such as atmospheric conditions and input parameters required to perform the ILP method contribute to the calibration differences."

*Line 104: "The data used" is referred to this work/paper? Sometimes I feel there is a lack of information to understand the sentences. The entire sentence is not clear to me.*

Rephrased from:

"The data used are from the period 2017-2021 in two locations, Davos (Switzerland) and Rome (Italy) in order to evaluate the ILP performance under different conditions."

to:

"In order to evaluate the ILP performance under different conditions, we used the sun photometer measurements from the 2017-2021 period at two locations: Davos (Switzerland) and Rome (Italy)."

*Line 109-110: Something is missing in this sentence.*

Rephrased from:

"For this study, we used the PFRN27 as reference in Davos (part of the PFR reference triad), while in Rome we used the PFRN14 (2017-2019) and PFRN01 (2021). We also used the co-located CIMEL in each campaign for cross-validation."

to:

"For this study, we used the sun photometer, PFRN27 (part of the PFR reference triad), as a reference in Davos (part of the PFR reference triad), while in Rome we used PFRN14 (2017 - 2019) and PFRN01 (2021). We also used the co-located CIMEL in each campaign for AOD cross-validation."

*Line 110: **a** co-located Cimel?*

Corrected.

*Line 111: I don't understand what "initial" means in this sentence. Please rephrase.*

Rephrased from:

"(one both in its initial and a later modified version)"

to:

"(one of the POM masters in two different versions due to modification between QUATRAM II and III to make it suitable for lunar measurements)"

*Table 1: The mention to the ability to perform lunar measurements is only mentioned in this table. Why not mention this in Sect. 2.1.2?*

Mentioned in the modification of the previous comment.

*Line 145: FWHM is different for 1640 nm spectral band. Please correct*

Corrected.

*Line 155: Please correct the typo in "Langley".*

Corrected.

*Lines 171-172: Is there a typo at the end of the sentence? I'm not able to understand it.*

They are 2 sentences (one ending at $\tau_g$ and one starting at $m_g$). We split the text in 3 to make it easier for the reader.

Rephrased from:

"Knowing the atmospheric pressure, we can calculate $\tau_R$ and the total column of gases absorbing at a certain wavelength we can calculate $\tau_g$. mg and ma are the air masses corresponding to gases and aerosols."

to:

"The value of $\tau_R$ is calculated using the atmospheric pressure. The value of $\tau_g$ is calculated from the total column of gases absorbing at a certain wavelength. The values of $m_g$ and $m_a$ are the air masses corresponding to gases and aerosols, respectively."

*Line 181: Please correct "to increase" and "their status".*

Corrected.

*Line 182: I consider it is more useful to include that it is usable at every type of station.*

Rephrased as proposed.

*Line 184: Please delete the extra point.*

Corrected.

*Lines 185-185: Some commas could improve the readability of this sentence.*

Modified from:

"Considering the Rayleigh scattering and gas absorption optical depths known, $\tau_a$. is the only required parameter to be retrieved before we calculate the calibration constant."

to:

"The Rayleigh scattering and gas absorption optical depths can be calculated, so AOD is the only parameter to be retrieved before deriving the calibration constant."

*Line 199: To retrieve…*

Corrected.

*Line 201: It is strange using "also" in a new paragraph…*

Deleted.

*Line 204: What is the "case" you are referring?*

To each different campaign or instrument. Replaced "case" with campaign.

*Lines 204-206: Please rephrase.*

Rephrased from:

"To evaluate ILP we calibrated the POMs using a PFR as a reference for each case. For measurements of DSI from co-located instruments at the same wavelength with I1 being the DSI at the ground measured from PFR, I2 the DSI measured from POM the same time, I01 the PFR calibration constant and I02 the POM calibration constant:"

to:

"To evaluate ILP, we calibrated the POMs using a PFR as a reference for campaign. Assume two co-located instruments (a PFR and a POM) measure DSI at the same wavelength. If $I_1$ is the DSI at the ground measured with a PFR, $I_2$ is the DSI measured with a POM at the same time, $I_{01}$ the PFR calibration constant and $I_{02}$ the POM calibration constant then the irradiances satisfy the following equation:"

*Line 211: Add "the" before POM.*

Corrected.

*Lines 211-212: Please rephrase: "The calibration constants and raw signals are in the units measured by each instrument and are corrected ..."*

Rephrased from:

"The calibration constants and raw signals are in the units each instrument measures and corrected for the Earth-Sun distance differences by shifting everything to 1 A.U."

to:

"The calibration constants and raw signals are in the instrument units (different for each instrument), and were also corrected for an Earth-Sun distance of 1 A.U."

*Line 214: A diurnal variation seems to be the reason for restricting ratios. Is it something that happens every day? If so, "A" is not necessary.*

Not necessarily every single day, but in most days it is present and quite consistent. Removed "A".

*Line 215: Time interval?*

Corrected.

*Line 216: Rest of data? This sentence seems unfinished...*

Replaced "rest" with "remaining data".

*Line 216: 2 std of the points? Please correct.*

Resolved in the next comment.

*Lines 216-217: I don't understand this sentence…*

Rephrased from:

"We removed all point calibrations outside 2 standard deviations of the points during each day in a loop until 2 standard deviations fall below or equal to 0.5% of the daily median calibration. If the remaining points are below 20, the day is rejected."

to:

"We checked whether the two standard deviations (σ) of all points during each day fell below or were equivalent to 0.5% of the daily median calibration. If the 2σ were above 0.5% of the daily calibration, we repeatedly removed all points outside the 2σ range until the day satisfied this criterion. If the remaining points of that day fell below 20 during this procedure, the day was rejected. "

*Lines 217-218: Is the same criterion than the stated in line 215?*

Not exactly, but the result is the same making the 20 measurement criterion unnecessarily mentioned in line 215. Removed from line 215.

*Line 218: Is this rejection regarding calibration error a visual analysis or it is based on a certain threshold?*

This procedure aimed at just removing visually daily calibration outliers that persisted after the automatic screening. It was dependent on the condition without pre-selected threshold. It refers to cases where the signal of one instrument was consistently unusually high or low during the day, which did not result in higher standard deviation and therefore the corresponding data satisfied the criterion of lines 216-217. Or for cases with abrupt changes of the calibration (see the reply on the comment about lines 407-409).

*Lines 223-224: I'm not sure I've understood correctly this sentence. Between 2 months the authors use a linear interpolation between two values, with a constant value during the whole month. But what happens in the case of the first month of measurements?*

For the first month of the measurements we use the same calibration for all measurements as explained in the line 223. Rephrased to make it easier for the reader. Modified:

 "We assumed that the monthly calibrations correspond to the last day of each month at 12:00 UTC. For measurements between 2 monthly calibrations, we use linear interpolation to calculate the calibration at the time of the measurement. For the first month of each campaign, we use the monthly calibration constant for all measurements of the month."

to:

"For the first month of each campaign, we used the monthly calibration constant for all measurements of the month. For the rest of the months, we assumed that the monthly calibrations correspond to the last day of each month at 12:00 UTC. For measurements between two monthly calibrations, we used linear interpolation to calculate the calibration at the time of the measurement. The interpolation is only based on these two consecutive monthly calibrations."

*Line 225: I suggest to remove this sentence: "The actual wavelength of each instrument may vary".*

Removed the sentence.

*Line 238: The authors are talking about two different datasets: original ESR and calculated from calibration transference. In this sentence, the authors are stating that the second dataset (own calibration) does not include NO2 correction but SKYNET (first dataset) includes it? Please explain why.*

The POM AOD retrieved with the ESR calibration is provided by the ESR retrieval method (includes NO2 correction), while the second dataset (from calibration transfer) by the GAW-PFR method, which does not include such correction. When comparing the second POM AOD dataset with the PFR we eliminate effects of post processing and calibration, while by comparing the original AOD datasets from each network we can see the overall effect of all differences between them. A separate comparison (supplement Fig. S1) shows purely the effect of the calibration. In this way we assess the effect of different factors.

However, the effect of NO2 in these particular wavelengths is minor. At 870 nm there is no effect. At 500 nm according to the AERONET data, the NO2 correction changes the AOD on average by approximately 0.0015 in Rome. In Izana (high altitude station) it is 0.0004-0.0006 (Cuevas et al.,2019).

Cuevas, E., Romero-Campos, P. M., Kouremeti, N., Kazadzis, S., Räisänen, P., García, R. D., Barreto, A., Guirado-Fuentes, C., Ramos, R., Toledano, C., Almansa, F., and Gröbner, J.: Aerosol optical depth comparison between GAW-PFR and AERONET-Cimel radiometers from long-term (2005–2015) 1 min synchronous measurements, Atmos. Meas. Tech., 12, 4309–4337, https://doi.org/10.5194/amt-12-4309-2019, 2019.

*Line 243: Please add reference for the AERONET algorithm.*

Added Smirnov et al., 2000 and Giles et al., 2019 for AERONET and Kazadzis et al., 2018b for the PFR cloud screening.

New references:

Smirnov, A., Holben, B., N., Eck, T., F., Dubovik, O., Slutsker, I.: Cloud-screening and quality control algorithms for the AERONET database, Remote Sens. Environ., 73.3, 337-349, https://doi.org/10.1016/S0034-4257(00)00109-7, 2000.

*Line 244: Are the authors using a double screening? AERONET & PFR cloud screenings.*

Each AOD dataset separately is cloud screened by one algorithm, except the POM datasets that are not cloud screened initially. Further cloud screening is accomplished through the synchronisation of datasets for the comparison (30 second difference threshold).

Therefore, the POM-PFR comparison is according to the PFR screening only. The CIMEL-PFR comparison includes double screened data (synchronous measurements of cloud screened CIMEL AOD according to AERONET and cloud screened PFR AOD according to the GAW-PFR algorithm).

*Line 252: I understand what the authors are pointing here, but don't you think that anticipating the reader with results which will be published later in the paper is not recommendable? Maybe the authors can reference a published paper (Campanelli, 2023) or move this section to the results section?*

Campanelli et al. 2023 is already cited in the sentence, so we just remove the reference to the section 3.1 of the present manuscript.

*Line 257: Is the first time that tau_sc is written as sc-AOD? I think it is better to standardize the terms to prevent the reader from getting lost when reading the paper.*

Homogenised such mentions and added to an abbreviation table in the appendix.

*Line 261: This is a general comment. Don't you think that the sentence "There are three parameters included in this section" has perfect meaning without the words "which be"? In my view, the use of these extra words leads to a wordy reading.*

Removed "which we".

*Line 264: Its variability? Variability of what variable?*

Of the SSA ("SSA value and variability").

*Line 267: Is there an estimation on the different uncertainty of levels 1.5 and 2.0 or this sentece is relating large uncertainty and the lack of QA/QC of AERONET level 1.5 data?*

The second. It clarifies that the level 1.5 is not quality assured. However, in the related AERONET document

(https://aeronet.gsfc.nasa.gov/new_web/Documents/U27_summary_final.pdf) it becomes clear that the uncertainty of SSA is larger for lower AODs. Since we use a lower AOD threshold compared to AERONET level 2.0 (due to lack of data otherwise) the SSA uncertainty should be larger.

*Line 268: change in? (Repeated twice in this sentence).*

Corrected.

*From line 268 onwards: From this point onward, this referee will no longer correct grammar-related aspects as it is not the object of the review and I am not a native English speaker. I would like to recommend that the authors undertake a thorough correction of the writing and the English used in this manuscript. It is difficult to understand some reasoning presented in this work.*

Answered in the response of the general comments.

*Line 277: Are these abbreviations already included before in the text? If so, please make use of abbreviations.*

We made use of the abbreviations.

*Line 288: Since the restriction to QUATRAM-II campaign is presented in this specific paragraph, focus on SVA, this referee understand that these input files are only referred to SVA and not to the rest of parameters (P, SA, etc). Can the authors confirm?*

No, it refers to all parameters for all the sub-studies described in section 2.4.2. For clarification we modify the text as following:

Instead of:

"The SVA is derived with the disk scan method, an on-site calibration procedure (Nakajima et al., 2020; Campanelli et al., 2023). To investigate the effect of these input files we performed a set of ILP calibrations under different conditions in 3 sub-studies. For this section, we used only data from QUATRAM II as it is the longest campaign"

we write:

"SVA is derived with the disk scan method, an on-site calibration procedure (Nakajima et al., 2020; Campanelli et al., 2023).

To investigate the effect of the aforementioned input parameters, we performed a set of ILP calibrations under different conditions in three sub-studies. For these sub-studies, we only used data from QUATRAM II as it was the longest campaign"

*Line 289: Since the authors will follow the same structure in section 3.2.2 I would recommend separate these sub-studies more clearly in this section, according to the sub-sub-sub-sections defined in section 3.2.2.*

Separated the section 2.4.2 to 2.4.2.1, 2.4.2.2 and 2.4.2.3 to follow the structure of 3.2.2.

*Line 299: Do the authors mean: in the case of SA? I don't understand what the authors are referring here.*

Both refractive index and surface albedo from AERONET data are not available at 340, 400 and 500 nm as explained in the previous 2 sentences. We refer to all three parameters (RRI, IRI and SA), but the selection for RI and SA one was done differently.

The 940 nm are not used by the ILP procedure or for AOD retrieval and remained in the text by mistake.

Rephrased from:

"For the rest of the wavelengths (340, 400 and 500 and 940 nm) we had to select values based on the existing wavelengths either by interpolation and extrapolation (we used linear) (RRI, IRI) or by a separate criterion (SA). The SA selection is based on the observed SA and the spectral dependence of the SA in the IGBP library from the LibRadtran package (Emde et al., 2016)."

to:

"For the rest of the wavelengths (340, 400 and 500 nm) we had to select values based on the existing wavelengths. For RRI and IRI, we used linear interpolation and extrapolation to estimate their values at those three missing wavelengths. The SA selection at 340, 400 and 500 nm is based on its observed values and its spectral dependence in the IGBP library from the LibRadtran package (Emde et al., 2016)."

*Line 324: The method uses log(DSI) plus Rayleigh and gas absorption terms versus m*AOD_sc, right?*

Yes. Corrected by mentioning the absorption terms after the logarithm of DSI.

*Line 346: The methods presented in this paper related to POM retrievals seem to be more accurate at high AOD conditions but then, cloud screening impose AOD to be below 0.4. I understand that this assumption is out of the scope of this paper, but I'm wondering if the authors could explain shortly in plain words if you see any inconsistency between these criteria or if you expect to change them in the future.*

The POM calibrations are expected to be more accurate at higher AOD due to the sky radiance inversions required to retrieve sc-AOD (Nakajima et al., 2020; Kudo et al., 2021 - already in the manuscript). However, when it comes to the observed systematic bias, we do not see this in the our results (section 3.2.1). We also have very few cases of AOD above 0.4 even in Rome, making it difficult to investigate the effect this criterion really has.

*Line 369: Yes, it is quite noticeable the differences observed during this campaign, especially at 500 nm. Are these results already published or the authors have any explanation for these high discrepancies?*

It's not perfectly clear to which differences the referee refers here.

The CIMEL-PFR AOD differences, which are mentioned in line 369 are generally small and within the uncertainties. During QUATRAM II at 500 nm, there was a median difference of almost 0.008 with the majority of the data within the WMO limits and the uncertainty of 0.01. This comparison shows larger differences compared to the other CIMEL-PFR comparisons, but since most data are within the uncertainties we don't consider the differences very large. It's unclear why this comparison is not as good as the others.

The POM-PFR AOD differences during QUATRAM II are indeed particularly large for 1 of the 2 instruments we used from that campaign (POMCNR) and the largest from all campaigns. This instrument at 500 nm in supplement Figure S1 shows the largest deviation between the POM-PFR comparison and the AOD differences purely attributed to the calibration. The reason for the additional AOD bias of POMCNR beyond the calibration effect (almost 0.01) is not clear. If it was due to the post processing differences, we should see this in the other campaigns/instruments. It could be due to some issue with the instruments' performance or solar tracking.

*Line 384: I don't understand "random differences within the retrieved uncertainty". Can the authors elaborate what this partial conclusion means?*

Here we mean that the overall contribution of the post processing algorithm and instrument technical differences between the networks to the AOD differences on average are very close to 0. Also, that the vast majority of the AOD differences (at least 90%) are within the PFR AOD uncertainties. However, the difference on average is not exactly 0, so strictly speaking the full sentence is incorrect. There may be biases that are just significantly smaller than the uncertainty and the standard deviation of the AOD differences (for example due to the NO2 correction effect at 500 nm).

Rephrased from:

"These results suggest that the post processing algorithm and instrument technical differences between the networks are a source of only random AOD differences within the retrieval uncertainty."

to:

"These results suggest that the overall contribution of the post-processing algorithm and instrument differences between the networks result in AOD differences that are within the PFR AOD retrieval uncertainty."

*Figure 1: What are the two red lines at ±0.01? It will be useful here include "(POM_IL-PFR)", as it is stated in the legend. The same for "POM_TR" and "Cimel-PFR".*

0.01 is the uncertainty of PFR and CIMEL AOD at air mass 1 (the uncertainty decreases as the air masses increase) at these wavelengths.

*Table 2: I suggest to define the instruments as POM-XXX or POM/XXX to make clear the two words (instrument + location). I know that you have also the difference in calibration method (POM_XX) that can cause confusion...*

The calibration method is not relevant in the case of Table 2 and more than 1 instrument is used for the same location. The names are chosen to be the same with Campanelli et al. 2023 for all common POMs in order to link better the 2 manuscripts and avoid confusion between them.

We changed the name of POM11 to POMSPZ for consistency of the name format.

*Lines 407-409: Please rephrase this sentence. The part in parenthesis is a sentence itself and it is very difficult to read as it is.*

Rephrased from "The values in the supplement show some minor differences compared to Campanelli et al., 2023 for some months mainly due differences in the day selection that are larger for August 2018 in Davos (where we observed an abrupt calibration shift during the month and removed the days before the shift as the monthly calibration is attributed to the end of the month when retrieving AOD). "

to:

"The values in the supplement show some minor differences compared to Campanelli et al., (2023) for some months mainly due differences in the selected days. The difference is larger for August 2018 in Davos. During this month we observed an abrupt shift of daily calibrations early in the month. Hence, we removed the days before the shift as the monthly calibration is attributed to the end of the month when retrieving the AOD."

*Line 413: Is it expected the uncertainties of ILP to be purely random?*

The differences that we discuss in this particular sentence can explained by the random component of the uncertainty and/or changes in the instrument's response. This fact doesn't exclude the existence of other components of uncertainty that are systematic, as we can see in different parts of the manuscript. That systematic bias though was an observational finding not something already predicted from a past theoretical study.

Rephrased from:

"The ILP calibrations show either positive and negative fluctuations for consecutive months in the same location between 0.17-2.3% with a median absolute value of 0.55% and a standard deviation of 0.87%. It can be attributed both to changes in the instruments and the random uncertainty of the ILP method."

to:

"The ILP calibrations show either positive or negative fluctuations for consecutive months at the same location lying in the 0.17-2.3% range with a median absolute value of 0.55% and a standard deviation of 0.87%. These calibration fluctuations can either be attributed to changes in the instruments' response or the random component of the ILP method uncertainty."

*Line 413: Is the estimation "evident"?*

Rephrased from:

 "An estimation of the uncertainty magnitude is evident in the coefficient of variation (CV%) of the daily ILP calibrations per month (Campanelli et al., 2023 preprint table 2a) which are between 0.18%-2.87% at 500 and 870 nm."

to:

 "The coefficient of variation (CV%) of the daily ILP calibrations per month (Campanelli et al., 2023 Table 2a) is an estimate of the ILP monthly calibration uncertainties. The CV% for the ILP calibrations used in this study lies in the 0.18%-2.87% range at 500 and 870 nm."

*Line 424: What are the fluctuations expressed here? Are authors referring to the amplitude of the uncertainties previously reported? I'm not able to understand this paragraph...*

There are differences between monthly calibrations for consecutive months. As "fluctuations of ILP" we refer to that month-to month variability of the ILP retrieved calibration constants. As "fluctuations of the transfer-based calibrations" the month-to month variability of the PFR-based calibration transfers to POMs.

The month-to-month variability of the 2 calibration methods does not coincide. For example, the ILP calibrations constant may change by 1% one month compared to the previous one with the calibration transfer for the same months being unchanged. In this part, we talk about this different behaviour.

The "the month-to-month fluctuations of their difference" is the month-to-month variability of the difference between the ILP calibration and the calibration transfer for the same instrument.

In another paragraph of the same section, we compare the above fluctuations with calibration uncertainties.

Rephrased from:

"The fluctuations of ILP and transfer-based calibrations do not coincide, which is reflected in the month-to-month fluctuations of their difference being 0.01%-1.93% with median absolute value of 0.55% and standard deviation 0.96%"

to:

"The month-to-month variability of the ILP method and transfer-based calibrations do not coincide. This is reflected in the month-to-month variability of the calibration differences between both methods, which is in the 0.01%-1.93% range. Their median absolute value is 0.55% and their standard deviation 0.96%"

*Line 449: Can you please help the reader with a reference to the corresponding figure, table or number?*

Added a reference after (ROM19).

*Figure 2: AOD in the figure caption is referred to PFR, right? Can you clarify?*

Yes. Added "PFR" in the caption.

*Line 466: Can you please help the reader with a reference to the corresponding figure, table or number? The same for the rest of partial conclusions.*

Added references as proposed.

*Line 471: Is 2.4.1. a reference to a Section?*

Yes. Added the word ("section").

*Figure 3: Is the SSA extracted from AERONET? Please clarify.*

Yes. Added the clarification.

*Line 481 and Figure 4: Is the AE retrieved from PFR? Please clarify.*

Yes. Added clarifications.

*Line 496: Is this analysis restricted to QUATRAM_II campaign as stated in line 289?*

Yes.

*Line 502: As commented in line 287, the name of these sub-studies should be similar and clearly related to the description in line 287.*

Resolved as explained in the comment for line 287.

*Line 514: Are the acronyms already defined? If so, use the acronym.*

Changed to the acronym.

*Line 518: Is not all this section referred to QUATRAM-II?*

Yes. Deleted the last sentence of this line.

*Line 530: What type of modifications the authors expect in the v4.2?*

After further testing we saw no effect from the selection criteria, so we removed the 'modification of the selection criteria'.

Currently, it's not clear what modification to the algorithm could prevent such computational instabilities.

*Figure 5: Do the authors think that the title of the graphs are needed? The same for the rest of figures. I suggest to include panels (a) and (b) rather than "left side" and "right side" in the caption. The same for the rest of figures.*

The titles offer some basic information on what the graph is about. In any case, we do not consider them undesirable.

Added panels a) and b).

*Table 4: The first sentence in the caption case seem unfinished. What "selected case" means?*

Rephrased from:

"The %difference between the original ILP and transferred calibrations minus the %difference between the ILP under selected conditions and the transferred."

to:

"The percentage difference between the original ILP and calibration transfer minus the percentage difference between the ILP method, for selected conditions and the calibration transfer."

'Selected' case is defined earlier. It is a name for a particular selection of input parameters (the ones that lead to higher calibration constant for all input parameters).

In the new manuscript structure the part of the manuscript where it is defined will be followed by the results, so the reader will not lose that information while reading several different sections in between.

*Figure 8: Again, comment on title and legend meaning.*

Resolved in the same way as the comment about figure 5.

*Line 615: I recall reading in the first part of the paper about the difference between Skyrad and MRI and the importance of including the second method. However, unfortunately, after so much information, at this point, the reader no longer remembers that information. This comment is not aimed at repeating the information in this section but simply to inform the authors that reading this paper, as it is presented, is quite challenging.*

Modified as described in the response of the general comments.

*Table 6: As in the case of the figures and previous tables, in the caption should appear the information as written in the table. In this case, I believe that (P5th, P95th) should be mentioned.*

Corrected as proposed.

*Table 7: The same for Delta V0.*

Resolved as in the previous comment.

*Line 662: Maybe it is interesting to provide here some numbers about the systematically lower AOD?*

Added the median differences of AOD.

*Line 681 onwards: Please, focus only on the most important information...*

Deleted: "In addition, AOD at 500 nm was above 0.1 in QUATRAM I (8/2017) and below 0.05 the AOD at 500 nm is above 0.1, while in in QUATRAM III (10/2021) below 0.05, but the calibration difference is was smaller in QUATRAM I. Similarly, in Rome at during QUATRAM II, the first month (5/2019) shows simultaneously exhibits the lowest AOD and SSA variability in at both wavelengths. At 500 nm, the second and fourth months (6 and 8/2019) show a smaller calibration difference, while AOD is higher and all three parameters are more variable."

*Line 693: Is it necessary to repeat again the acronyms?*

Changed to the acronyms.

*Line 745: Can the authors provided here some numbers when they talk about the underestimation?*

Added the range of calibration differences % and median AOD differences.